# Approximation, Estimation and Optimization Errors for a Deep Neural Network

**B. Keene**                                                                    *keene@ucf.edu*
*Department of Mathematics*
*University of Central Florida*
*Orlando, FL 32816, USA*

**G. Welper**                                                                *gerrit.welper@ucf.edu*
*Department of Mathematics*
*University of Central Florida*
*Orlando, FL 32816, USA*

**Reviewed on OpenReview:** *https://openreview.net/forum?id=dzND5haNvA*

## Abstract

The error of supervised learning is typically split into three components: approximation, estimation and optimization errors. While all three have been extensively studied in the literature, a unified treatment is less frequent, in part because of conflicting assumptions. Current approximation results rely on carefully hand crafted weights or practically unavailable information, which are difficult to achieve by gradient descent. Optimization theory is best understood in over-parametrized regimes with more weights than samples, while classical estimation errors require the opposite regime with more samples than weights. This paper contains two results which bound all three error components simultaneously for (nonconvex) training of the second but last layer of deep fully connected networks on the unit sphere. The first uses a regular least squares loss and shows convergence in the underparametrized regime. The second uses a kernel based loss function and shows convergence in both under and over-parametrized regimes.

## 1 Introduction

In this paper, we consider supervised learning of fully connected neural networks without bias: For network $f_\theta$ with weights $\theta$ and normalized training samples $(x_i, y_i)$ on the $d$-dimensional sphere, we minimize the loss

$$\ell(\theta) = \frac{1}{2N} \sum_{i=1}^{N} |f_\theta(x_i) - y_i|^2$$

by gradient descent. We also consider alternative losses, which allow more flexibility with regard to the number of samples and network size. The main results provide a complete error analysis including *approximation errors*, *estimation errors* and *optimization errors* (Shalev-Shwartz & Ben-David, 2014).

### 1.1 Literature Review

**Approximation Error:** If the data points $y_i = f(x_i)$ are generated by some unknown target function $f$, how well can the network approximate it, i.e. how large is the error $\inf_\theta \|f_\theta - f\|_{L_2(D)}$ on some domain $D$, ignoring error contributions from sampling and optimization algorithms? Typical results establish bounds

$$\inf_\theta \|f_\theta - f\|_{L_2(D)} \le cn(\theta)^{-r}, \qquad\qquad f \in K, \qquad\qquad (1)$$

for some rate $r > 0$, with an asymptotic rate $n(\theta)^{-r}$, where $n(\theta)$ is a complexity measure of the network, e.g. width, depth or total number of weights. Any quantifiable rate requires some prior conditions on $f$, here given by membership in a compact set $K$, which typically consists of functions with bounded Sobolev, Besov, Barron or other smoothness norms.

First results show universal approximation properties (Cybenko, 1989; Hornik et al., 1989; Barron, 1993; Zhou, 2020; Lu et al., 2017; Hanin & Sellke, 2017) and reproduction of classical approximation rates for targets with Sobolev and Besov regularity (Gribonval et al., 2022; Gühring et al., 2020; Opschoor et al., 2020; Li et al., 2019; Suzuki, 2019). More recent papers provide super-convergence results, where networks outperform classical methods, as well as optimality benchmarks like manifold width, for the price of discontinuous weight assignments (Yarotsky, 2017; 2018; Yarotsky & Zhevnerchuk, 2020; Daubechies et al., 2022; Shen et al., 2019; Lu et al., 2021). Approximation results with smoothness requirements more tailored to neural networks use Barron and related spaces (Bach, 2017; Klusowski & Barron, 2018; Weinan et al., 2022; Li et al., 2020; Siegel & Xu, 2020; 2022a; Bresler & Nagaraj, 2020). Several surveys are given in (Pinkus, 1999; DeVore et al., 2021; Weinan et al., 2020; Berner et al., 2022).

In the above results, the weights are not trained by practical optimizers, but rather hand-picked or sampled from practically unknown distributions. Practical networks do not always achieve the theoretical bounds Adcock & Dexter (2021); Grohs & Voigtlaender (2023) The papers Jentzen & Riekert (2022); Ibragimov et al. (2022); Gentile & Welper (2022); Welper (2024b;a) are closely related to the results in this paper and provide an analysis of approximation errors in combination with gradient descent training. Finally Siegel & Xu (2022b); Siegel et al. (2023); Beck et al. (2022); Herrmann et al. (2022) consider approximation or estimation with alternative optimizers like greedy algorithms.

**Generalization and Estimation errors:** Practically, one can neither evaluate nor optimize the $L_2(D)$ error directly, and therefore trains the sample or empirical loss, resulting in the empirical risk minimizer $\hat{\theta}$. The resulting expected loss is called the *generalization error* and split into the *approximation error* and a remaining *estimation error*, (Shalev-Shwartz & Ben-David, 2014):

$$\|f_{\hat{\theta}} - f\|_{L_2(D)}^2 = \inf_{\theta} \|f_{\theta} - f\|_{L_2(D)}^2 + \left( \|f_{\hat{\theta}} - f\|_{L_2(D)}^2 - \inf_{\theta} \|f_{\theta} - f\|_{L_2(D)}^2 \right).$$

There are many techniques to bound generalization and estimation errors, for example by establishing uniform bounds between the expected and sample loss

$$\sup_{\theta} \left| \|f_{\theta} - f\|_{L_2(D)}^2 - \frac{1}{2N} \sum_{i=1}^{N} |f_{\theta}(x_i) - f(x_i)|^2 \right| \lesssim \mathcal{C} + N^{-1/2} \tag{2}$$

for some complexity measure $\mathcal{C}$ of the neural networks like VC-dimension or Rademacher complexity. VC-dimension bounds of the form VC-dim $\lesssim \tilde{\mathcal{O}}(Ln(\theta))$ for total number of weights $n(\theta)$ and depth $L$ are in Neyshabur et al. (2017); Bartlett et al. (1998); Harvey et al. (2017). Bounds for the Rademacher complexity tend to be independent of the size of the network as e.g. $\mathcal{O}\left( \frac{\sqrt{L} \prod_{\ell=1}^{L} \|W^{\ell}\|_F}{\sqrt{N}} \right)$ for weight $W^{\ell}$ in the Frobenius norm from Golowich et al. (2018). Similar bounds are in Neyshabur et al. (2015); Liang et al. (2019); Tu et al. (2020). While the norm $\|W^{\ell}\|_F$ may grow for wide networks with standard scaling, newer results depend on the difference $\|W^{\ell} - W_0^{\ell}\|$ to some reference or initial weights $W_0^{\ell}$ in Frobenius and other matrix norms, which tends to be small in the over-parametrized limit and leads to some combined generalization and gradient descent convergence results (Cao & Gu, 2020). Other Rademacher complexity bounds rely on smoothness (Weinan et al., 2022; 2019). Further techniques include margin theory (Jakubovitz et al., 2019; Neyshabur et al., 2018; Bartlett et al., 2017) mutual information (Asadi et al., 2018; Steinke & Zakynthinou, 2020) compression (Arora et al., 2018) and Besov regularity (Suzuki, 2019). Empirical observations show that against conventional wisdom neural networks generalize well in over-parametrized regimes, with enough weights to fit random data, (Neyshabur et al., 2017; Zhang et al., 2017; Geiger et al., 2019).

**Optimization Error:** Both the empirical loss minimizer $\hat{\theta} = \operatorname{argmin} \ell(\theta)$ and the approximation error $\inf_{\theta} \|f_{\theta} - f\|_{L_2(D)}^2$ build on an optimization problem. Since these are non-convex, can we practically compute

the minimizers, or at least a sufficiently good substitute? This question is addressed in the optimization literature for neural networks. The approach in this paper relies on the neural tangent kernel (NTK) coined in Jacot et al. (2018) and introduced simultaneously in Li & Liang (2018); Allen-Zhu et al. (2019); Du et al. (2019b;a). The concept is further developed in Zou et al. (2020); Arora et al. (2019a;b); Su & Yang (2019); Lee et al. (2019); Song & Yang (2019); Zou & Gu (2019); Kawaguchi & Huang (2019); Chizat et al. (2019); Oymak & Soltanolkotabi (2020); Ji & Telgarsky (2020); Nguyen & Mondelli (2020); Bai & Lee (2020); Chen et al. (2021); Song et al. (2021); Lee et al. (2022); Gentile & Welper (2022); Welper (2024b;a). In this paper we use lower bounds for the NTK that originate from Bietti & Mairal (2019); Geifman et al. (2020); Ji et al. (2020); Chen & Xu (2021). The optimization literature contains may other approaches that we only mention briefly: Landscape analysis (Nguyen & Hein, 2017; Ge et al., 2018; Du & Lee, 2018; Soltanolkotabi et al., 2019; Venturi et al., 2019), Wasserstein gradient flow (Soudry & Carmon, 2016; Safran & Shamir, 2018; Chizat & Bach, 2018; Mei et al., 2018; Rotskoff & Vanden-Eijnden, 2018; Sirignano & Spiliopoulos, 2020), as well as several overviews (Weinan et al., 2020; Berner et al., 2022; Roberts et al., 2022).

**Generalization and Optimization:** The definition of stochastic gradient descent intertwines optimization with sampling and the literature provides positive effects on generalization. This can be analyzed in abstract settings, e.g. the paper Hardt et al. (2016) bounds the generalization error between empirical and expected risk of SGD iterates of general convex and non-convex objective functions. In addition, it bounds the expected loss between SGD and empirical risk minimizers in the convex case.

Other papers are more specific to neural networks, i.e. Cao & Gu (2020); Nitanda & Suzuki (2021) consider a combination of estimation and optimization in over-parametrized regimes. The latter proves generalization errors for stochastic gradient descent by comparing its dynamics with an NTK idealization, yielding minimax optimal SGD convergence rates. In addition, it does not require any lower bounds on the NTK eigenvalues, similar to Welper (2024b) and the current paper. The papers Wang & Ma (2023); Liu et al. (2022); Park et al. (2022); Neu et al. (2021) find improved estimation bounds by explicitly incorporating gradient descent dynamics. Another closely related set of results is Drews & Kohler (2022); Kohler & Krzyzak (2022), which consider all three error contributions and control the optimization error based on the contributions of the final layer.

## 1.2 New Contributions

This paper contains generalization error bounds of the form

$$\|f_\theta - f\|^2_{L_2(\mathbb{S}^{d-1})} \lesssim m^{-a} + m^b N^{-c},$$

where $m$ is the width of the fully connected network $f_\theta$, $N$ is the number of samples and $a, b, c \geq 0$ are constants specified more closely in the main Theorems 2.2, 2.4. The weights $\theta$ on the left hand side are the output of gradient descent training on two different discrete losses. For simplicity, we confine the data to uniform samples on the unit sphere and train only the non-convex second but last layer. The new contributions of this bound are as follows:

**Less Over-Parametrization:** While over-parametrization in *total number of weights* is common in practical neural networks, current optimization theory heavily relies on much larger networks with more *width* than samples $m \gg N$. In contrast, successful architectures like AlexNet Krizhevsky et al. (2012) or ResNet He et al. (2016) are much thinner $m < N$ (including channels, width and height).

Our result does not require any relation between $m$ and $N$. For $b > 0$, this provides small bounds in $m < N$ regimes (including under-parametrization). For a kernel based loss, Theorem 2.4 achieves $b = 0$, resulting in small bounds for any regime $m \lesseqgtr N$ (including under- and over-parametrization). To the best of our knowledge, this is the first result that shows gradient descent convergence for deep neural networks independent of the relation between network size and number of samples.

**Unified Analysis:** The given bound contains approximation errors ($m^{-a}$), estimation errors ($m^b N^c$) and optimization errors ($\theta$ is trained by GD). These bounds can be found in the literature in isolation, but

rarely in one single theorem because of conflicting assumptions. The optimization literature requires over-parametrization so that all data $y_i = f(x_i)$ can be correctly reproduced by the network $y_i = f_\theta(x_i)$. This directly entails maximally large Rademacher complexity and hence conflicts with common assumptions for estimation error bounds. In the infinite sample limit considered in approximation theory, this also confines the target functions $f$ to neural networks themselves, instead of rich classes of practical relevance. Therefore, the approximation literature ignores the optimization problem in favour of simpler hand-picked weights.

Only a few papers in the literature can balance all required assumptions and provide bounds for approximation, estimation and optimization in one single theorem. The papers Drews & Kohler (2022); Kohler & Krzyzak (2022), use non-standard architectures, which allow a derivation of error bounds from the last layer alone. This is a convex training objective, in contrast to this paper, which analyzes the training contributions form non-convex layers of standard fully connected networks. The papers Cao & Gu (2020); Nitanda & Suzuki (2021) include estimation and optimization errors, but require excessive over-parametrization as is typical in the current literature.

**Estimation:** The prior work Gentile & Welper (2022); Welper (2024b;a) establishes approximation and optimization error bounds. This paper contributes a corresponding analysis of the estimation errors, based on two alternative approaches. The first follows more traditional lines and bounds the complexity of the network directly. The second relies on smoothness bounds that arise from gradient descent training.

### 1.3 Overview

We use the following two techniques to show estimation errors.

**Network Complexity:** The most common complexity measures are VC-dimension and Rademacher complexity, which in turn can be bounded by chaining techniques, i.e. by Dudley's inequality (Shalev-Shwartz & Ben-David, 2014). In our case, it is convenient to skip the Rademacher complexity and use Dudley's inequality directly because it has already been used to establish NTK concentration inequalities for the approximation and optimization error bounds (Welper, 2024b).

We minimizes the sample loss

$$\ell(\theta) = \frac{1}{2} \sum_{i=1}^{N} |f_\theta(x_i) - f(x_i)|^2 \tag{3}$$

for $N$ uniformly random normalized samples $x_i$ on the unit sphere $\mathbb{S}^{d-1}$ with gradient descent. We show that as long as the error does not satisfy the approximation and estimation error estimate

$$\|f_\theta - f\|_{L_2(\mathbb{S}^{d-1})}^2 \lesssim m^{-a} + m^b N^{-c},$$

with network width $m$, the gradient descent error decreases exponentially. The rates $a$, $b$ and $c$ are specified in Theorem 2.2 below and depend on the Sobolev smoothness of the target function $f$. Although we optimize the discrete sample loss, we bound the error in the continuous $L_2(\mathbb{S}^{d-1})$ norm and therefore obtain the expected or generalization error. The result is comparable to standard machine learning theory. In particular, the second term requires that the number of samples $N$ is larger than the width $m$ of the network (up to some power).

**Smoothness:** While requiring more samples $N$ than width $m$ matches common wisdom in machine learning theory, it does not explain the empirical observation that neural networks generalize well in over-parametrized regimes. To establish generalization error bounds in this regime, we rely on a different complexity measure: The approximation and optimization results in Welper (2024b;a) establish that the Sobolev norm of the gradient descent iterates $\|f_{\theta^n} - f\|_{H^s(\mathbb{S}^{d-1})}$ remains uniformly bounded, independent of the size of the network. If $s > 1 + d/2$, Sobolev embedding theorems imply that $f_\theta$ is uniformly Lipschitz and therefore the estimation error bound (2) can be proven by uniform laws of large numbers (Vershynin, 2018, Section 8.2).

Unfortunately, the current theory only provides bounds for $s < 1/2$, insufficient for the argument. We may, however, proceed with the *kernel loss*

$$\ell^k(\theta) = \frac{1}{2N} \sum_{i=1}^{N} \langle k(x_i, \cdot), f_\theta - f \rangle^2, \tag{4}$$

with uniformly random $x_i$, which probes the residual $f_\theta - f$ with an integral kernel $k(x,y)$, $x,y \in \mathbb{S}^{d-1}$ in the $L_2$ inner product $\langle \cdot, \cdot \rangle$ and is easier to bound in low regularity settings. Moreover, for common kernels like the heat kernel, Gaussian kernel $e^{-|x-y|^2/\sigma^2}$ or Laplacian kernel $e^{-|x-y|/\sigma}$, this loss converges to the standard mean squared loss (3) for $\sigma \to 0$ and proper normalization.

Although our interest in this kernel loss is of theoretical nature, to explore new arguments for generalization in over-parametrized regimes, it is similar to variational losses in VPINNs (Kharazmi et al., 2019; 2021), used to solve PDEs with neural networks. In this application, it is common that PDE solutions do not admit continuous point evaluations, and instead one probes the residual $\langle f_\theta - f, v \rangle$ with test functions $v$ from some linear subspace, for which the given kernels would be one example.

The kernel loss also bears a resemblance with randomized smoothing (Cohen et al., 2019): In order to mitigate adversarial attacks on a classifier $f_\theta$ for $\mathcal{Y}$ classes, these methods choose the class that is most likely under normal perturbations $\epsilon \sim \mathcal{N}(0, \delta^2)$

$$g(x) = \arg\max_{c \in \mathcal{Y}} \Pr\left[f_\theta(x + \epsilon) = c\right].$$

In comparison, for a Gaussian kernel with variance $\delta^2$, the kernel loss is identical to the mean squares loss of the averaged network

$$g_\theta(x) = \mathbb{E}\left[f_\theta(x + \epsilon)\right] = \langle k(x, \cdot), f_\theta \rangle.$$

The second main result shows that for the kernel loss, gradient descent decreases exponentially until it reaches the approximation and estimation error

$$\|f_\theta - f\|_{L_2(\mathbb{S}^{d-1})}^2 \lesssim m^{-a} + N^{-c},$$

again with rates $a$ and $c$ dependent on the Sobolev regularity of $f$ as specified in Theorem 2.4. The two error contributions on the left hand side are decoupled and we achieve the worst case of the approximation error $m^{-a}$ and the sample error $N^{-c}$. Contrary to the first result and conventional machine learning theory, this allows meaningful generalization errors even in over-parametrized regimes with more samples $N$ than width $m$.

**Estimation and Gradient Descent:** To obtain the results, we do not bound the difference between empirical and expected loss (2) directly. Instead, we compare the gradient descent evolution to an idealized method trained on the continuous $L_2$ loss:

$$\theta^n: \qquad \text{trained by gradient descent on loss (3) or (4).}$$
$$\bar{\theta}^n: \qquad \text{trained by gradient descent on loss } \frac{1}{2}\|f_{\bar{\theta}} - f\|_{L_2(\mathbb{S}^{d-1})}^2.$$

Convergence of the latter is established in Welper (2024a). From this we prove convergence for the former based on perturbation analysis and sample errors for the respective *gradients* (not the loss as in standard analysis (2)). This approach is reminiscent of Cohen et al. (2002), which analyzes adaptive PDE solvers by comparing them with idealized infinite dimensional ones. The sample errors are established by Dudley's inequality for the sample loss (3) and by matrix Bernstein inequalities for the kernel loss (4).

## 2 Main Results

This section contains the main results of the paper.

### 2.1 Setup

The setup is almost identical to Welper (2024b;a), with the major difference that the references train on an idealized continuous $L_2(\mathbb{S}^{d-1})$ loss, whereas we train on practical sample losses.

**Notations** We denote generic constants by $c$, which may be different in each occurrence, but do not depend on the width $m$, number of samples $N$ or input dimension $d$. Alternatively, we use the shorthand $a \lesssim b$, $a \gtrsim b$, $a \sim b$ to denote $a \leq cb$, $a \geq cb$, $a \lesssim b \lesssim a$, respectively.

For integer $s$, Sobolev spaces $H^s(\mathbb{S}^{d-1})$ consist of all functions on $\mathbb{S}^{d-1}$ with $L_2(\mathbb{S}^{d-1})$ bounded weak derivatives of order $s$. For non-integer $s$, these spaces can be defined by the decay of their expansion

$$\|f\|_{H^\alpha(\mathbb{S}^{d-1})}^2 = \sum_{l=0}^{\infty} \sum_{j=1}^{\nu(l)} \left(1 + l^{1/2}(l + d - 2)^{1/2}\right)^{2\alpha} \left|\left\langle Y_l^j, f \right\rangle\right|^2 \tag{5}$$

in spherical harmonics

$$Y_\ell^j, \qquad\qquad \ell = 0, 1, 2, \ldots, \qquad\qquad 1 \leq j \leq \nu(\ell), \tag{6}$$

for suitable numbers $\nu(\ell)$, see e.g. Barceló et al. (2020). We denote the corresponding inner product by $\langle \cdot, \cdot \rangle_{H^s(\mathbb{S}^{d-1})}$.

**Network** We consider fully connected networks without bias

$$
\begin{aligned}
f^1(x) &= W^0 x, \\
f^{\ell+1}(x) &= W^\ell m_\ell^{-1/2} \sigma\left(f^\ell(x)\right), \quad \ell = 1, \ldots, L
\end{aligned}
\tag{7}
$$

of depth $L$, with normalized inputs in the unit sphere $x \in D := \mathbb{S}^{d-1}$ and standard scaling. We summarize all trainable weights in the parameter $\theta$ and abbreviate the network by $f_\theta(x) := f^{L+1}(x)$. To obtain a simple non-convex model problem, we optimize the second but last layer $W^{L-1}$ and initialize all weights randomly

$$
\begin{array}{llll}
W^L \in \{-1, +1\}^{1 \times m_L} & \text{i.i.d. Rademacher} & \text{not trained,} \\
W^{L-1} \in \mathbb{R}^{m_L \times m_{L-1}}, & \text{i.i.d. } \mathcal{N}(0,1) & \text{trained} \\
W^\ell \in \mathbb{R}^{m_{\ell+1} \times m_\ell}, \ell = 1, \ldots, L-2 & \text{i.i.d. } \mathcal{N}(0,1) & \text{not trained} \\
W^0 \in \mathbb{R}^{m_1 \times d}, & \text{i.i.d. } \mathcal{N}(0,1) & \text{not trained.}
\end{array}
$$

All hidden layers are of comparable size, the input $d$-dimensional and the output scalar:

$$m := m_{L-1}, \qquad\qquad 1 = m_{L+1} \leq m_L \sim \cdots \sim m_1 \geq d.$$

**Activation Functions** We require smooth activation functions with no more than linear growth and derivatives bounded as follows:

$$|\sigma(x)| \lesssim |x|, \qquad |\sigma^{(i)}(x)| \lesssim 1 \qquad i = 1, 2, \qquad |\sigma^{(j)}(x)| \leq p(x), \qquad j = 3, 4, \tag{8}$$

for some polynomial $p(x)$.

**Training** All networks are trained by gradient descent

$$\theta^{n+1} = \theta^n - \gamma \nabla_\theta \ell(\theta^n), \tag{9}$$

with learning rate $\gamma > 0$. We use different losses $\ell(\theta)$ for the main results and define them in the respective sections.

**Neural Tangent Kernel**  The main results require coercivity of the neural tangent kernel, which has been shown in Welper (2024b) for ReLU activations based on Bietti & Mairal (2019); Geifman et al. (2020); Chen & Xu (2021), but remains open for smoother activations (8) used in this paper. Since we only train the second but last layer, in our case the *neural tangent kernel (NTK)* is informally defined as

$$\Gamma(x,y) = \lim_{\text{width}\to\infty} \sum_{r=1}^{R} \partial_r f_\theta(x) \partial_r f_\theta(y), \tag{10}$$

with partial derivatives $\partial_{W_{ij}^{L-1}}$ abbreviated by a single index $\partial_r$ with $r = 1, \ldots, R := m_L m_{L-1}$. The coercivity condition is then stated as

$$\langle f, Hf \rangle_{H^S(\mathbb{S}^{d-1})} \gtrsim \|f\|_{H^{S-\beta}(\mathbb{S}^{d-1})}, \qquad\qquad Hf := \int_D \Gamma(\cdot, y) f(y) \, dy \tag{11}$$

for some $\beta > 0$, all $S \in \{0, s\}$, some smoothness level $0 \le s \le \frac{\beta}{2}$ and all $f \in H^s(\mathbb{S}^{d-1})$. Again, for networks with ReLU activations, bias and all layers trained this is true with $\beta = d/2$, see Welper (2024b).

In addition, the main results require

$$\Sigma^k(1) \neq 0, \qquad\qquad k = 1, \ldots, L, \tag{12}$$

for the Gaussian process that describes the forward evaluation of the random initial network in the infinite width limit (Jacot et al., 2018). Its correlation matrices $\Sigma(x, y) = \Sigma(x^T y)$ only depend on the angle between $x, y \in \mathbb{S}^{d-1}$ and are, defined by

$$\Sigma^{\ell+1}(x, y) := \mathbb{E}_{u,v\sim\mathcal{N}(0,A)} \left[ \sigma\left(u\right), \sigma\left(v\right) \right], \qquad A = \begin{bmatrix} \Sigma^\ell(x, x) & \Sigma^\ell(x, y) \\ \Sigma^\ell(y, x) & \Sigma^\ell(y, y) \end{bmatrix}, \qquad \Sigma^0(x, y) = x^T y,$$

As for coercivity, this property is known for ReLU activations, where $\Sigma(1) = 1$, see Chen & Xu (2021), and is expected to be a minor technical assumption for smoother activations (8). The condition is directly related to the NTK coercivity and with it left for future work.

## 2.2  Result I: Pointwise Sampling

For the first result, we use the standard least squares loss

$$\ell(\theta) = \frac{1}{2} \frac{1}{N} \sum_{i=1}^{N} [f_\theta(x_i) - f(x_i)]^2, \tag{13}$$

with $N$ independent uniform samples $x_i \in \mathbb{S}^{d-1}$. We first collect all major assumptions.

**Assumption 2.1.** *Assume:*

1. *The neural network (7) - (8) is trained by gradient descent (9).*

2. *The NTK satisfies coercivity (11) for $0 \le 2s \le \beta$ and the forward process satisfies (12).*

3. *All hidden layers are of similar size: $m_0 \sim \cdots \sim m_{L-1} =: m$.*

4. *Smoothness is bounded by $0 < s < 1/2$.*

5. *Define $h$ and $\tau$ as follows and choose learning rate $\gamma$ and an arbitrary $\alpha$ so that*

$$h = c_h m^{-\frac{1}{2}\frac{1}{1+\alpha}}, \qquad \tau = h^{2\alpha} m, \qquad \gamma \lesssim h\sqrt{m}, \qquad 0 \le \alpha < 1 - s.$$

*for some constant $c_h$ that may depend on the initial error $\|f_{\theta^0} - f\|_{L_2(\mathbb{S}^{d-1})}$.*

The following result is similar to Welper (2024a, Theorem 2.2), which only considers approximation and optimization errors. While the reference trains on the continuous $L_2(\mathbb{S}^{d-1})$ loss, we train on the discrete sample loss and therefore also include estimation errors.

**Theorem 2.2.** *Assume we train the sample loss* (13), *let Assumption 2.1 be satisfied, let* $\|f\|_{L_\infty(\mathbb{S}^{d-1})} \lesssim m^{1/2}$, *and define*

$$\Delta_{sample}(m, N) = c_\Delta \frac{m^{3/2}}{N^{1/2}} h^{1-\frac{\alpha s}{2\beta}} \|f_{\theta^0} - f\|_{H^s(\mathbb{S}^{d-1})}^{-1}$$

*for some sufficiently large* $c_\Delta$. *Then with residual* $\kappa^n := f_{\theta^n} - f$ *and probability at least* $1 - cL(e^{-m} + e^{-\tau})$, *while the gradient descent error exceeds the final approximation and estimation error*

$$\|\kappa^k\|_{L_2(\mathbb{S}^{d-1})}^2 \geq c_a \left( m^{-\frac{1}{2}\frac{\alpha}{1+\alpha}} + \Delta_{sample}(m, N) \right)^{\frac{s}{\beta}} \|\kappa^0\|_{H^s(\mathbb{S}^{d-1})}^2, \qquad k < n, \qquad (14)$$

*we have*

$$\|\kappa^n\|_{L_2(\mathbb{S}^{d-1})}^2 \leq Ce^{-\gamma[h^\alpha + \Delta_{sample}(m,N)]n} \|\kappa^0\|_{L_2(\mathbb{S}^{d-1})}^2, \qquad \|\kappa^n\|_{H^s(\mathbb{S}^{d-1})}^2 \leq C\|\kappa^0\|_{H^s(\mathbb{S}^{d-1})}^2.$$

*for sufficiently large constants* $c_a$, $c$ *and* $C$ *independent of* $m$, $\kappa^0$ *and* $\kappa^n$.

The proof is in Section B.2. The assumptions relate the smoothness, the size of the network and number of samples. The only major assumption is the coercivity (11), (12), which is open for our activations but can be easily inferred from the literature (Bietti & Mairal, 2019; Geifman et al., 2020; Chen & Xu, 2021) for ReLU activations. In the latter case, $\beta$ depends on the input dimension $d$ and therefore all other bounds are also dimension dependent, although this is not explicit in the stated results. See Welper (2024b) for details, and numerical verification for smoother activations required in the theorem.

The result shows that gradient descent converges exponentially fast until the error is sufficiently small (14) and we have

$$\|\kappa^n\|_{L_2(\mathbb{S}^{d-1})}^2 < c_a \left( m^{-\frac{1}{2}\frac{\alpha}{1+\alpha}} + \Delta_{sample}(m, N) \right)^{\frac{s}{\beta}} \|\kappa^0\|_{H^s(\mathbb{S}^{d-1})}^2, \qquad (15)$$

The first summand, $m^{-\frac{1}{2}\frac{\alpha}{1+\alpha}}$ together with the smoothness $\|\kappa^0\|_{H^s(\mathbb{S}^{d-1})}^2$ provides a typical approximation error bound of the form (1). The second term $\Delta_{sample}$, bounds the sample error and has the typical $N^{-1/2}$ dependence on the number of samples together with some factors of $m$ and $h$ that measure the complexity of the network. In particular, for the overall error to be small, we must have more samples $N$ than width $m$.

**Remark 2.3.** *We can simplify the final error bound* (15) *as follows. First, we allow all constants to depend on the initial value* $\kappa^0$ *and drop all terms that contain it. Next, we distribute the outer exponent* $s/\beta$ *to the summands, for the price of a slightly worse constant, and obtain*

$$\|\kappa^n\|_{L_2(\mathbb{S}^{d-1})}^2 \lesssim m^{-\frac{1}{2}\frac{\alpha}{1+\alpha}\frac{s}{\beta}} + \Delta_{sample}(m, N)^{\frac{s}{\beta}}.$$

*Next, we unravel the definitions of* $\Delta_{sample}$ *and* $h$ *to arrive at*

$$\|\kappa^n\|_{L_2(\mathbb{S}^{d-1})}^2 \lesssim m^{-\frac{1}{2}\frac{\alpha}{1+\alpha}\frac{s}{\beta}} + \left( \frac{m^{3/2}}{N^{1/2}} h^{1-\frac{\alpha s}{2\beta}} \right)^{\frac{s}{\beta}} \lesssim m^{-\frac{1}{2}\frac{\alpha}{1+\alpha}\frac{s}{\beta}} + \left( \frac{m^{3/2}}{N^{1/2}} \left[ m^{-\frac{1}{2}\frac{1}{1+\alpha}} \right]^{1-\frac{\alpha s}{2\beta}} \right)^{\frac{s}{\beta}}$$

*Observing the constraints on* $s$, $\beta$ *and* $\alpha$, *we abbreviate the exponents by some numbers* $a, b, c \geq 0$ *to obtain*

$$\|f_\theta - f\|_{L_2(\mathbb{S}^{d-1})}^2 \lesssim m^{-a} + m^b N^{-c}.$$

## 2.3 Result II: Kernel Sampling

**Motivation**  Generalization errors can be derived from bounds of the form

$$\sup_\theta \left| \|f_\theta - f\|_{L_2(\mathbb{S}^{d-1})} - \frac{1}{2N} \sum_{i=1}^N |f_\theta(x_i) - f(x_i)| \right| \leq \mathcal{C} + N^{-1/2},$$

which controls the difference between the expected and empirical loss, uniformly in all parameters $\theta$ contained in a set $\Theta$ of all relevant parameters. The bound on the right hand side depends on some complexity measure $\mathcal{C}$ such as Rademacher complexity. By the supremum in the estimate, the complexity bound usually depends on the size of the hypothesis class $\{f_\theta \mid \theta \in \Theta\}$. In Theorem 2.2, this gives rise to the $m$ dependent $\Delta_{sample}(m, N)$ and therefore to the requirement of more samples than network size (although we technically apply the argument to the gradient, not the loss).

In order to decouple the size of the network form the number of samples, we use the observation from Welper (2024b) or Theorem 2.2 that in the initial NTK regime the Sobolev norm $H^s(\mathbb{S}^{d-1})$ of the residual $\kappa := f_\theta - f$ does not grow, i.e. $\|\kappa^n\|_{H^s(\mathbb{S}^{d-1})} \le \|\kappa^0\|_{H^s(\mathbb{S}^{d-1})}$. Hence, we may replace the sample error with a bound of the form

$$\sup_{\|\kappa\|_{H^s(\mathbb{S}^{d-1})} \le \|\kappa^0\|_{H^s(\mathbb{S}^{d-1})}} \left| \|\kappa\|^2_{L_2(\mathbb{S}^{d-1})} - \frac{1}{2N} \sum_{i=1}^N |\kappa(x_i)|^2 \right| \le \mathcal{C} + N^{-1/2}, \tag{16}$$

which is *independent of the network size*. If $s$ is sufficiently large so that $H^s(\mathbb{S}^{d-1})$ is embedded into $L^\infty(\mathbb{S}^{d-1})$ with some margin, this estimate can be shown by a uniform law of large numbers, as e.g. in Vershynin (2018, Chapter 8.2). Such an embedding also ensures a bounded loss function, which is assumed for standard VC dimension and Rademacher complexity bounds of the generalization error. For small $s$, a favorable right hand side cannot be expected. Indeed, the map $\kappa \to \|\kappa\|^2_* := \frac{1}{2N} \sum_{i=1}^N |\kappa(x_i)|^2$ from $H^s$ to $\mathbb{R}$ must be continuous, otherwise one can find a perturbation $\tilde{\kappa}$ so that $\|\kappa\|_{L_2} - \|\tilde{\kappa}\|_{L_2} \le \|\kappa - \tilde{\kappa}\|_{L_2} \le \|\kappa - \tilde{\kappa}\|_{H^s}$ is small and $\|\kappa\|_* - \|\tilde{\kappa}\|_*$ is large and as a result the left hand side of (16) must be large for either $\kappa$ or $\tilde{\kappa}$. By the Sobolev embedding theorem (Adams & Fournier, 2008) the continuity holds for $s \ge d/2$ and fails otherwise. Unfortunately, our results provide low Sobolev regularity $s < 1/2$, insufficient for this embedding.

For a first theoretical exploration of alternative complexity measures and in order to stay compatible with earlier approximation and optimization results, we change the point evaluation to localized integrals

$$\ell^k(\theta) := \frac{1}{2N} \sum_{i=1}^N \langle k(x_i, \cdot), f_\theta - f \rangle^2, \tag{17}$$

with uniformly random $x_i$ and some integral kernel $k \colon \mathbb{S}^{d-1} \times \mathbb{S}^{d-1} \to \mathbb{R}$ that is smoothing and allows continuous evaluation for all $s > 0$. Note that many standard kernels, like heat, Gaussian and Laplacian kernels, converge to the Dirac delta for their "width" going to zero. As a result, the loss $\ell^k$ converges to the regular mean square loss $\ell$ in (13). See Section 2.4 for more details.

**Kernels** Before we state the main result, we need some properties of the kernel. First, we assume it is zonal, i.e. that $k(x, y) = k(x^T y)$. As a result, by the Funk-Hecke formula (Atkinson & Han, 2012) the eigenfunctions are spherical harmonics $Y_l^j$ (6) and for the corresponding eigenvalues $\lambda_{lj}$, we require

$$\begin{aligned} 1 \lesssim \lambda_{lj} \lesssim 1, \quad & l \le L, \quad 1 \le j \le \nu(l), \\ \lambda_{lj} \lesssim 1, \quad & l > L, \quad 1 \le j \le \nu(l), \end{aligned} \tag{18}$$

so that up to a limiting level $L > 0$ the eigenvalues are of unit size and falling thereafter. In addition, the kernels are bounded

$$\sup_{x \in D} \|k(x, \cdot)\|_{L_2(\mathbb{S}^{d-1})} \le C_k. \tag{19}$$

We defer a more thorough discussion of kernels with the given properties to Section 2.4 and for the time being consider a convolutional kernel on the line instead of zonal kernel on the sphere, to avoid technicalities. In this case, a natural example is the Gaussian kernel $k_G(x - y) = \frac{1}{\sqrt{2\pi}t} e^{-\frac{|x-y|^2}{t^2}}$. It is diagonalized by the Fourier transform with continuous eigenvalues

$$\lambda(\omega) = \hat{k}_G(\omega) = \frac{1}{2\sqrt{\pi}} e^{-\frac{\omega^2 t^2}{4}}$$

for which we obtain (18) by the simple observation

$$\lambda(\omega) \lesssim 1, \qquad \omega \in \mathbb{R}, \qquad \lambda(\omega) \gtrsim 1, \qquad \omega^2 t^2 \le 1 \Leftrightarrow |\omega| \le \frac{1}{t},$$

Similar kernels on the sphere are more technical and discussed in Section 2.4.

**Result**  Unlike more traditional complexity measures in machine learning, the smoothness $\|\kappa^n\|_{H^s(\mathbb{S}^{d-1})} \leq \|\kappa^0\|_{H^s(\mathbb{S}^{d-1})}$ is a byproduct of the gradient descent method and *independent of the size of the network*. This yields error bounds with decoupled approximation and sampling error in the following theorem.

**Theorem 2.4.** *Assume we train the kernel loss* (17) *with conditions* (18), (19) *and corresponding constants $C_k$ and $L$. Let Assumption 2.1 be satisfied and for arbitrary $\tau_N \lesssim N$ define*

$$\Delta_{sample}(m, N) = c_\Delta \left[ C_k^2 \left( \frac{\tau_N}{N} \right)^{1/2} + C_k^{-2} \left( \frac{N}{\tau_N} \right)^{1/2} L^{-s} + L^{-s} \right].$$

*for some sufficiently large $c_\Delta$. Then with $\kappa^n := f_{\theta^n} - f$ and probability at least $1 - ce^{-\tau} - 2\tau_N \left[ e^{\tau_N} - \tau_N - 1 \right]^{-1}$ while the gradient descent error exceeds the final approximation and estimation error*

$$\|\kappa^k\|_{L_2(\mathbb{S}^{d-1})}^2 \geq c_a \left( m^{-\frac{1}{2}\frac{\alpha}{1+\alpha}} + \Delta_{sample}(m, N) \right)^{\frac{s}{\beta}} \|\kappa^0\|_{H^s(\mathbb{S}^{d-1})}^2, \qquad k < n,$$

*we have*

$$\|\kappa^n\|_{L_2(\mathbb{S}^{d-1})}^2 \leq Ce^{-\gamma[h^\alpha + \Delta_{sample}(m,N)]n} \|\kappa^0\|_{L_2(\mathbb{S}^{d-1})}^2, \qquad \|\kappa^n\|_{H^s(\mathbb{S}^{d-1})}^2 \leq C\|\kappa^0\|_{H^s(\mathbb{S}^{d-1})}^2.$$

*for sufficiently large constants $c_a$, $c$ and $C$ independent of $m$, $\kappa^0$ and $\kappa^n$.*

As for Theorem 2.2 the error decays exponentially until the final sum of approximation and sample error is reached. Unlike Theorem 2.2, the sample error $\Delta_{sample}(m, N)$ does not depend on the network size $m$. Hence, the final error is the worse of the approximation and the sample error and provides meaningful error bounds both in under- and over-parametrized regimes.

If we choose the best possible ratio $\frac{N}{\tau_N} = C_k^4 L^s$ between the number of samples $N$ and the success probability parameter $\tau_N$, given the parameters $L$ and $C_k$ of the kernel, we obtain

$$\Delta_{sample}(m, N) \leq (1 + c_\Delta)L^{-s/2}, \tag{20}$$

converging to zero for large $L$ corresponding to locally concentrated kernels, see Section 2.4.

The theorem contains the inequality $\|\kappa^n\|_{H^s(\mathbb{S}^{d-1})}^2 \leq C\|\kappa^0\|_{H^s(\mathbb{S}^{d-1})}^2$ and therefore $\kappa^n = f_{\theta^n} - f$ remains bounded in the Sobolev norm. Hence, for the generalization error, we consider the hypothesis class of bounded Sobolev functions, instead of neural networks with bounded weights as in typical Rademacher or margin bounds.

**Remark 2.5.** *Analogous to Remark 2.3, we can obtain a simplified error bound after training*

$$\|f_\theta - f\|_{L_2(\mathbb{S}^{d-1})}^2 \lesssim m^{-a} + N^{-c}.$$

*Here, we use that $\Delta_{sample}$ does not depend on $m$ and can be bounded by* (20). *To eliminate $L$, we can choose e.g. $\tau_N = \sqrt{N}$ and solve the ratio $\frac{N}{\tau_N} = C_k^4 L^s$ for $N$. Note, however, that $C_k$ and $L$ are not independent, which is worked out for the heat kernel in* (22), *below.*

## 2.4  Kernels

In this section, we consider kernels that meet our assumptions (18) and (19).

**Heat Kernel**  We first consider the *heat kernel* $k_t(x, y)$ on the sphere $\mathbb{S}^{d-1}$, defined as the solution of the heat equation with Dirac delta as initial condition (Zhao & Song, 2018)

$$\partial_t k_t(\cdot, y) - \Delta k_t(\cdot, y) = 0, \qquad\qquad k_0(\cdot, y) = \delta(\cdot, y),$$

where $\Delta$ is the Laplace-Beltrami operator on the sphere and $\delta(x,y)$ the Dirac delta distribution on the sphere. On the flat space $\mathbb{R}^d$, this kernel is identical to the Gaussian kernel $(2\pi)^{-1/2}t^{-1}e^{-|x-y|^2/t^2}$, while on the sphere they differ. We use the heat kernel because it allows a particularly simple verification of our kernel assumptions.

Indeed, since the Laplace-Beltrami operator's eigenfunctions are spherical harmonics $Y_l^j$ with eigenvalues $-l(l+d-2)$ (Atkinson & Han, 2012), the heat equation has the explicit solution

$$k_t(x,y) = \sum_{l,j} e^{-l(l+d-2)t} Y_l^j(x) \left\langle Y_l^j, \delta(\cdot,y) \right\rangle = \sum_{l,j} e^{-l(l+d-2)t} Y_l^j(x) Y_l^j(y) \tag{21}$$

in its eigenbasis. Therefore, the eigenvalues of the kernel are $\lambda_{lj} = e^{-l(l+d-2)t}$ and with $l(l+d-2)t \leq 1 \Leftrightarrow l \lesssim t^{-1/2}$ we have

$$e^{-1} \leq \lambda_{lj} \leq 1, \qquad\qquad l \lesssim t^{-1/2},$$
$$\lambda_{lj} \leq 1, \qquad\qquad l \gtrsim t^{-1/2}$$

so that the kernel assumption (18) is satisfied with $L = ct^{-1/2}$. Since the eigenvalues decay exponentially, the Sobolev norms of the kernel are bounded. More concretely, Lemma D.1 in the supplementary material shows that

$$\|k_t(\cdot,y)\|_{L_2(\mathbb{S}^{d-1})}^2 \leq C_k^2 =: ct^{-d+3/2}.$$

In conclusion, the heat kernel satisfies all assumptions of Theorem 2.4 and the sample error (20) simplifies to

$$\Delta_{sample}(m,N) \leq (1+c_\Delta)t^{s/4}, \qquad\qquad \frac{N}{\tau_N} \sim t^{-\frac{1}{2}s-4d+6} \tag{22}$$

for the given number of samples. By construction, for $t \to 0$ the kernel converges to the Dirac delta and therefore the kernel loss (17) converges to the sample loss (13).

**Gaussian and Laplace Kernels**  In order to obtain some further insight into permissible kernels, we consider the Gaussian and Laplacian kernels

$$k_G(x-y) = \frac{1}{\sqrt{2\pi}t} e^{-\frac{|x-y|^2}{t^2}}, \qquad\qquad k_L(x-y) = \frac{1}{2t} e^{-\frac{|x-y|}{t}}$$

on the real line $\mathbb{R}$. For $s < 1/2$, these are clearly bounded in $H^s(\mathbb{R})$. Moreover, since these are convolutional kernels, the eigenvalues correspond to the Fourier coefficients, given by

$$\hat{k}_G(\omega) = \frac{1}{2\sqrt{\pi}} e^{-\frac{\omega^2 t^2}{4}}, \qquad\qquad \hat{k}_L(\omega) = \sqrt{\frac{2}{\pi}} \frac{1}{1+\omega^2 t^2}.$$

Thus, we easily obtain the analogues of the eigenvalue bounds (18):

$$\hat{k}_G(\omega) \lesssim 1, \qquad \omega \in \mathbb{R}, \qquad \hat{k}_G(\omega) \gtrsim 1, \qquad \omega^2 t^2 \leq 1 \Leftrightarrow |\omega| \leq \frac{1}{t},$$
$$\hat{k}_L(\omega) \lesssim 1, \qquad \omega \in \mathbb{R}, \qquad \hat{k}_L(\omega) \gtrsim 1, \qquad \omega^2 t^2 \leq 1 \Leftrightarrow |\omega| \leq \frac{1}{t}.$$

Similar results on the sphere are significantly more involved and beyond the scope of this paper. See e.g. Geifman et al. (2020, Appendix C) for an analysis of the Laplace kernel, without the fine grained dependence on $t$ required for our purposes.

## 2.5 Sketch of Proof

**Overview**  This section contains a short overview over the proofs of Theorems 2.2 and 2.4. We start by introducing a scale of continuous loss functions

$$\ell_S(\theta) := \frac{1}{2}\|f_\theta - f\|_{H^S(D)}^2, \qquad\qquad S \in \{0,s\}.$$

For $S = 0$, this loss is the generalization or $L_2(D)$ error. For $S = s > 0$, the loss is used to control the Sobolev smoothness of the neural networks later in the proof. For convenience, we abbreviate $\langle \cdot, \cdot \rangle_s := \langle \cdot, \cdot \rangle_{H^s(D)}$ and $\kappa := f_\theta - f$. A simple calculation and the mean value theorem yield that the loss evolves as

$$\ell_S(\theta^{n+1}) - \ell_S(\theta^n) = -\gamma \sum_r \langle \kappa, \partial_r f_{\bar{\theta}} \rangle_S \partial_r \ell(\theta^n),$$

for some $\bar{\theta}$ on the line segment between $\theta^n$ and $\theta^{n+1}$. On the left hand side, we use the $\ell_S$ loss, i.e. the generalization error or the smoothness, because these are the quantities we ultimately want to control in the theorems. On the right hand side, we have the discrete loss $\ell$ that we actually train on. By a perturbation argument, we replace the discrete loss with the continuous one:

$$\ell_S(\theta^{n+1}) - \ell_S(\theta^n) = -\gamma \sum_r \langle \kappa^n, \partial_r f_{\bar{\theta}} \rangle_S \partial_r \ell_0(\theta^n) - \gamma \sum_r \langle \kappa^n, \partial_r f_{\bar{\theta}} \rangle_S \left[ \partial_r \ell(\theta^n) - \partial_r \ell_0(\theta^n) \right].$$

$$=: \text{GD}_0 + \text{PERTURBATION}$$

The perturbation term will be bounded by

$$\text{PERTURBATION} \le \gamma \Delta_{sample}(m, N) \|\kappa^k\|_0 \|\kappa^k\|_S,$$

giving rise to the terms $\Delta_{sample}(m, N)$ in the main theorems.

**Convergence of** $\text{GD}_0$   We first consider the case that the perturbation term PERTURBATION is zero. This corresponds to training on the continuous $L_2(D)$ loss $\ell_0$ directly and has been studied in Welper (2024b;a). The convergence analysis is based on the observation that

$$\begin{aligned}
\ell_0(\theta^{n+1}) - \ell_0(\theta^n) &\le -\gamma \langle \kappa^n, H\kappa^n \rangle_0 + \text{perturbation}, \\
\ell_s(\theta^{n+1}) - \ell_s(\theta^n) &\le -\gamma \langle \kappa^n, H\kappa^n \rangle_s + \text{perturbation},
\end{aligned} \tag{23}$$

where $H$ is the linear integral operator induced by the neural tangent kernel

$$\Gamma(x, y) = \lim_{\text{width} \to \infty} \sum_{r=1}^{R} \partial_r f_{\theta^0}(x) \partial_r f_{\theta^0}(y).$$

In short, the terms $-\gamma \langle \kappa^n, H\kappa^n \rangle$ are a linearization of the gradient $-\gamma \|\nabla_\theta f_\theta\|^2$, usually found in gradient descent analysis. This linearization remains accurate because of the crucial observation that the weights $\theta^0 - \theta^n$ do not move far from their initial during training.

Ignoring the linearization error, the left hand sides of (23) are bilinear forms and therefore allow simple convergence proofs if the eigenvalues of $H$ are lower bounded. While this is true in over-parametrized regimes, for the $L_2(D)$ loss, the NTK is a compact operator and the eigenvalues converge to zero. Therefore, we consider a coupled evolution of the $\ell_0$ and $\ell_s$ losses: The latter ensures that the networks remain uniformly bounded in the Sobolev norms $H^s(D)$. This implies that the residual $\kappa^n$ is concentrated in low frequencies or equivalently eigenspaces corresponding to large eigenvalues. Therefore, the system is comparable to ones with lower bounded eigenvalues and allows us to prove convergence of the $\ell_0$ loss.

Since this theory already contains perturbation terms, the new terms PERTURBATION from sampling the gradient can be added to the theory with only minimal changes, shown in Theorem A.1.

**Bounds for** PERTURBATION   The main new contribution of the paper is to show that PERTURBATION, i.e. the difference between the *gradients* of the continuous loss $\ell_0$ and the discrete loss $\ell$ remain small. The arguments are similar to standard generalization error bounds, which address the difference between $\ell_0 - \ell$ directly.

1. Classical techniques to bound generalization errors include VC-dimension or Rademacher complexity, which, in turn, are sometimes bounded by Dudley's inequality (Shalev-Shwartz & Ben-David, 2014).

In Section B, we use the latter directly to bound PERTURBATION for the least squares loss (13). This is convenient because similar techniques are used in the prior work Welper (2024b) to show concentration of the NTK. The argument is reminiscent of the uniform law of large numbers as shown in Vershynin (2018, Chapter 8.2).

2. For the kernel loss (17), we take a different route in Section C. First, in the limit of infinite samples, heat, Gauss and Laplace kernels do not converge to the $L_2(D)$-loss. They rather converge to a dual norm $\|f_\theta - f\|_{H'}$ for primal norm $\|\cdot\|_H = \|\cdot\|_{L_2(D)}^2 + t\|\cdot\|_{H^\alpha(D)}^2$, some $\alpha > 0$ and a parameter $t$ for the width of the kernel.

   In the NTK convergence analysis, we have already shown that the residual $f_{\theta^n} - f$ is concentrated in low frequencies and as a result for carefully chosen $t$, the $L_2(D)$ part of the above norms dominate the $H^\alpha(D)$ part. Together with concentration results for the kernels, this time shown by matrix Bernstein inequalities, this implies gradient descent convergence in the $L_2(D)$ norm.

**Remark on Generalization Errors and Optimization**   Many generalization error bounds in the literature involve terms like $N^{-1/2}\|W^\ell - W_0^\ell\|_F$ (or other norms) that include the distance of the weights from their initial, which stays small in NTK gradient descent convergence proofs. Concretely, if we use the bound from the first equation in the proof of Theorem A.1, which underlies all main results, we obtain on the trained layer

$$\|W^{L-1} - W_0^{L-1}\|_F \leq m^{1/2}\|W^{L-1} - W_0^{L-1}\| =: m\|\theta - \theta_0\|_* \leq mh = m^{1 - \frac{1}{2}\frac{1}{1+\alpha}} \leq m^{1/2}$$

for some $\alpha \geq 0$, where $\|\cdot\|_F$ is the Frobenius norm, $\|\cdot\|$ the spectral norm and the middle equality the definition of the $\|\cdot\|_*$ norm. Therefore, the generalization error bounds can be further bounded by $N^{-1/2}\|W^{L-1} - W_0^{L-1}\|_F \leq N^{-1/2}m^{1/2}$. Hence, to obtain small generalization error, we must be in an under-parametrized regime with $m \leq N$. This conflicts with over-parametrization assumptions in most NTK convergence results. While the generalization bounds and NTK convergence may be reconciled with a finer analysis, we rely on the NTK results in Welper (2024b;a), which also work in under-parametrized regimes.

## 2.6   Extensions

This section contains a brief outlook on two major assumptions.

**Training of all Layers**   In the main Theorems 2.2 and 2.4, we train only the second but last layer, while all other weights remain frozen at their random initialization.

Including the remaining layers is relatively simple in standard NTK theory. The training of each layer adds one matrix $H^\ell$ to the NTK, so that training select layers $\Lambda \subset \{1, \ldots, L\}$ the gradient descent loss reduction in (23) becomes

$$\ell_0(\theta^{n+1}) - \ell_0(\theta^n) \leq -\gamma \sum_{\ell \in \Lambda} \langle \kappa^n, H^\ell \kappa^n \rangle_0 + \text{perturbation}.$$

For finite networks and their infinite width limits, the terms $\langle \kappa^n, H^\ell \kappa^n \rangle_0 \geq 0$ are non-negative so that including more layers in the training gives strictly sharper results.

The argument is more delicate for loss $\ell_s(\theta^n)$ with smoother $s > 0$, that we include in (23) to extend standard NTK theory to under-parametrized regimes. In this case, the terms $\langle \kappa^n, H^\ell \kappa^n \rangle_s$ are expected to be non-negative in the infinite width limit and only approximately so for finite width networks. Therefore, a careful additional perturbation and concentration analysis would be required to include the training of all layers in the main theorems, which is beyond the scope of this paper.

**Sample distribution**   The fully connected networks in (7) do not have a bias and as a result, the input $x = 0$ always yields output $f_\theta(x) = 0$, independent of the network weights. This degenerate case is avoided by restricting the input to the sphere. Introducing a bias would likely avoid this issue and allow more general domains, however, this would require a revision of cited results in Appendix D. Non-uniform distributions

would likely require some changes, in particular, the Sobolev norms in the smoothness bounds would need to be adapted accordingly.

## 3 Numerical Experiments

In this section, we supplement the theoretical findings with some preliminary numerical convergence rates with respect to network width and number of samples. We consider the following test case:

- *Network Architecture:* Fully connected with bias and ReLU activation.

- *Input data:* Uniformly distributed on the cube $[-1, 1]^d$.

- *Labels:* Gaussian density function of the input samples, i.e. $y_i = e^{-|x_i|^2/2}$.

- *Test Loss:* In order to approach the $L_2(D)$ error, the test loss is computed on a large number of 1000 uniformly sampled $x_i$ with mean squared loss, no matter the loss function used for training.

- *Kernel Loss:* To approximate the kernel loss, we uniformly sample $N$ points $z_i \in [-1, 1]^d$ and then for each $i$ we sample $N_k = 10$ samples from the normal distribution $\mathcal{N}(z_i, 0.01)$ to obtain $x_{ij} \in \mathbb{R}^{N \times N_k}$. Then, we approximate the kernel by Monte Carlo integration

$$\ell^k(\theta) \approx \sum_{i=1}^{N} \left[ \sum_{j=1}^{N_k} f_\theta(x_{ij}) - f(x_{ij}) \right]^2.$$

- *Training:* 20000 gradient descent steps with learning rate 0.05.

- *Repetition:* Since the randomness of data and initialization is important for the theoretical results, all reported losses are an average of 20 trials.

After sufficiently many gradient descent steps, in $m$ and $N$, Theorems 2.2 and 2.4 guarantee the convergence rates

$$\|f_\theta - f\|^2_{L_2(\mathbb{S}^{d-1})} \le m^{-0.0277} + m^{0.1991} N^{-0.0833},$$
$$\|f_\theta - f\|^2_{L_2(\mathbb{S}^{d-1})} \le m^{-0.0277} + N^{-0.0033},$$

respectively, where we have chosen dimension $d = 3$, $\beta = d/2$, which is true for ReLU activations as described after (11), $s = \min\{1/2, \beta/2\} = 1/2$, which is the maximal allowed value in the theorems and $\alpha = 1 - s$, which is the (excluded) upper limit. In addition, for the kernel bound, we use (22), with $\tau_N = N/2$, to balance the number of samples and the kernel width.

For comparison, we also consider the best possible rates for functions with bounded Sobolev norm $H^s(\mathbb{S}^{d-1})$. The best approximation rate can be computed with manifold width (Lorentz et al., 1996) and the best possible sampling rate is the minimax rate (Yang, 1999; Tsybakov, 2009), given by

$$m^{-\frac{s}{d}}, \qquad\qquad\qquad N^{-\frac{s}{2s+d}},$$

respectively. These are higher than the guaranteed rates in the main theorems. For classical approximation methods, the smoothness $s$ is capped by the polynomial's degree, which would be two for ReLU activations, yielding rate $m^{-2/3}$. The definition of the minimax rate is independent of the neural network and therefore allows $s \to \infty$ and hence rate $N^{-1/2}$ (up to possibly growing constants).

Figure 1 and Table 1 contain the estimated convergence rates of the test loss for training with mean squared loss (MSE) (13) and kernel loss (17). All plots are in log scale so that slopes along the $x$ or $y$ axes correspond to convergence rates. To compare the rates, note that the MSE loss $\|f_\theta - f\|^2_{L_2}$ reported in the tables differers from the error $\|f_\theta - f\|_{L_2}$ in the theoretical bounds by a square.

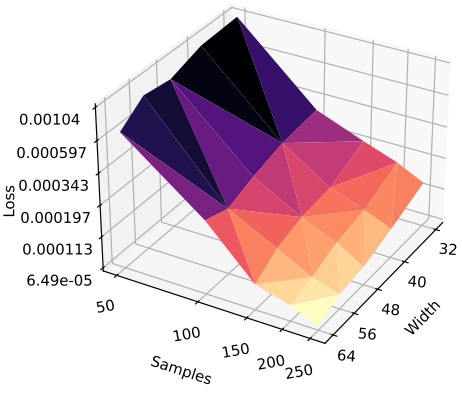

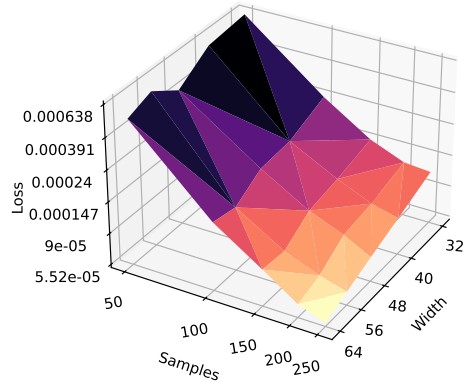

(a) MSE Loss, dim=3, depth=5.    (b) Kernel Loss, dim=3, depth=5.

Figure 1: Test loss for training with mean squared loss (13) (left) and kernel loss (17) (right). All axes are log-scaled so that the slope corresponds to convergence rates.

| | **Dimension 3 and Depth 5 – Trained with MSE Loss** | | | | | | | |
|---|---|---|---|---|---|---|---|---|
| | **dof rate** | | | | **$N$ rate** | | | |
| $m/\boldsymbol{N}$ | 100 | 150 | 200 | 250 | 100 | 150 | 200 | 250 |
| 40 | 0.331 | 1.33 | 1.33 | 1.19 | 1.9 | 1.36 | 0.743 | 0.896 |
| 48 | 0.833 | 0.328 | 0.628 | 1.06 | 1.86 | 1.13 | 0.933 | 1.25 |
| 56 | 1.13 | 0.969 | 1.25 | 0.434 | 2.2 | 1.07 | 1.08 | 0.684 |
| 64 | -1.03 | 2.05 | 0.967 | 1.47 | 1.56 | 2.08 | 0.582 | 0.985 |

| | **Dimension 3 and Depth 5 – Trained with Kernel Loss** | | | | | | | |
|---|---|---|---|---|---|---|---|---|
| | **dof rate** | | | | **$N$ rate** | | | |
| $m/\boldsymbol{N}$ | 100 | 150 | 200 | 250 | 100 | 150 | 200 | 250 |
| 40 | 1.08 | 0.676 | 0.748 | 1.07 | 1.65 | 0.939 | 0.797 | 0.482 |
| 48 | 0.229 | 0.517 | 0.433 | 1.05 | 1.23 | 1.07 | 0.744 | 0.987 |
| 56 | 1.69 | 1.77 | 1.75 | 1.11 | 2.02 | 1.1 | 0.734 | 0.539 |
| 64 | -0.54 | -0.218 | 0.898 | 1.19 | 1.71 | 1.21 | 1.25 | 0.713 |

Table 1: Estimated convergence rates between neighbouring losses for the given $m/N$. Left: Rate along the column, i.e. with respect to $m$. Right: Rate along rows, i.e. with respect to number of samples $N$. The first table is trained with mean squared loss (MSE) (13) and the second with kernel loss (17).

We first observe that the reported rates have significant variance despite being averaged over 20 separate runs. Individual runs are even more noisy. The majority of the table entries remain below the theoretically optimal rates, but are higher than the rates guaranteed by the theorems. While the theorems seem pessimistic, it is worth noting that the optimal rates utilize the high smoothness of the target function $s \geq 2$, whereas the theorems only allow smoothness up to $s < 1/2$. At the expense of high rates, this does allow much rougher target functions for which the optimal rates would be lower, as well. More detailed experiments are left for future work.

Finally note that despite severe over-parametrization, the loss decreases with respect to $N$, although with slowing rate. This matches the theoretical results for the kernel loss, which provides bounds in over-parametrized regimes, but does not decrease below the approximation error. More detailed results, including the loss and shallow networks, are contained in Appendix E.

## 4 Conclusion

The literature contains a large number of papers that bound approximation, estimation and optimization errors of neural networks. Because of conflicting assumptions, these results are usually not compatible. This paper contains a unified analysis that bounds all three error components at once. Unlike the contemporary literature, it does not rely on excessively wide networks or over-parametrization. Instead, we have seen for kernel loss that the generalization error can be bounded by the maximum of approximation and estimation errors, independent of any relation between width $m$ and number of samples $N$.

The generalization errors of this paper are achieved for a bounded number of gradient descent steps, where the training dynamics is dominated by the neural tangent kernel (NTK). Beyond this bound, the NTK loses its relevance and more nonlinear behaviour is possible. A rigorous analysis is left for future work.

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

# A Gradient Descent Convergence

## A.1 Convergence Result

In this section, we prove an abstracted convergence result that is the foundation for Theorems 2.2 and 2.4. To this end, let $\Theta \subset \mathbb{R}^R$ be a set of admissible weights, and $\mathcal{H}^s$, $s \in \mathbb{R}$ a set of Hilbert spaces. Then we consider

$$f_{\cdot} \colon \Theta \to \mathcal{H}^0, \qquad\qquad \theta \to f_\theta, \qquad\qquad (24)$$

$$\ell_s \colon \mathcal{H}^s \to \mathbb{R}, \qquad\qquad \theta \to \ell_s(\theta) =: \frac{1}{2}\|f_\theta - f\|_s^2, \qquad\qquad (25)$$

$$\ell \colon \mathcal{H}^0 \to \mathbb{R}, \qquad\qquad \theta \to \ell(\theta), \qquad\qquad (26)$$

where $\ell_0$ corresponds to the continuous loss, or generalization error, and $\ell$ to the discrete loss. The Hilbert spaces have norms $\|\cdot\|_s = \|\cdot\|_{\mathcal{H}^s}$ and are related by the interpolation inequality

$$\|\cdot\|_b \lesssim \|\cdot\|_a^{\frac{c-b}{c-a}} \|\cdot\|_c^{\frac{b-a}{c-a}} \qquad\qquad (27)$$

for for all $a, b, c \in \mathbb{R}$. Typically, we choose Sobolev spaces $\mathcal{H}^s = H^s(D)$, the neural network $\theta \to f_\theta(\cdot) \in L_2(D) = \mathcal{H}^0$, and some discrete loss $\ell$, but this is not important throughout this section. The statement of the following result relies on the empirical NTK and NTK

$$H_{\theta,\bar{\theta}} := \sum_{r=1}^R (\partial_r f_\theta)(\partial_r f_{\bar{\theta}})^*, \qquad\qquad H := \lim_{\text{width}\to\infty} \sum_{r=1}^R \partial_r f_{\theta^0} \partial_r f_{\theta^0}^*,$$

where $f^*$ is the $\mathcal{H}^0$-adjoint of $f$. Note that the adjoint, contained in the dual space $(\mathcal{H}^0)'$, applied to a function $v$ is $f^*(v) = \langle f, v \rangle$ and as a result, $H$ corresponds to the integral operator induced by the integral kernel in (11).

The following convergence result is almost identical to Welper (2024a, Theorem 3.1), up to a new error term for the difference between the Hilbert space loss $\ell_0$ and the discrete sample loss $\ell$ in (33).

**Theorem A.1.** *Assume we minimize the loss $\ell$ of the parametrized function $\theta \to f_\theta \in \mathcal{H}^0$ in (24), with gradient descent (9) and Hilbert spaces (27). Define the residual $\kappa^k = f_{\theta^k} - f$. Let $m$ be an indicator for the network size that satisfies the inequalities below. With $\alpha > 0$ from (31) below, assume that*

1. *$H$ is coercive for $S = 0$ and $S = s$ and some $\beta > 2s > 0$*

$$\|v\|_{S-\beta}^2 \lesssim \langle v, Hv \rangle_S, \qquad\qquad v \in \mathcal{H}^{S-\beta}. \qquad\qquad (28)$$

2. *For some norm $\|\cdot\|_*$, the distance of the weights from their initial value is bounded by*

$$\left\|\theta^k - \theta^0\right\|_* \lesssim 1,\; k = 1, \ldots, n \qquad \Rightarrow \qquad \left\|\theta^{n+1} - \theta^0\right\|_* \lesssim \frac{\gamma}{\sqrt{m}} \sum_{k=0}^n \|\kappa^k\|_0. \qquad (29)$$

3. *The learning rate $\gamma$ is sufficiently small so that*

$$\gamma \left\|\nabla_\theta \ell(\theta^n)\right\|_* \lesssim c_h m^{-\frac{1}{2}\frac{1}{1+\alpha}} =: h \qquad\qquad (30)$$

   *for some constant $c_h$ that may depend on the initial error $\|\kappa^0\|_0$.*

4. *For $S = 0$ and $S = s$, initial value $\theta^0$, any $\bar{\theta}, \tilde{\theta} \in \Theta$ and any $\bar{h} > 0$, the bounds $\left\|\theta^0 - \bar{\theta}\right\|_* \le \bar{h}$ and $\left\|\theta^0 - \tilde{\theta}\right\|_* \le \bar{h}$ imply*

$$\|H_{\tilde{\theta},\theta^0} - H_{\tilde{\theta},\bar{\theta}}\|_{S,0} \le c\bar{h}^\alpha, \qquad\qquad \|H_{\theta^0,\tilde{\theta}} - H_{\bar{\theta},\tilde{\theta}}\|_{S,0} \le c\bar{h}^\alpha. \qquad (31)$$

5. *For $S = 0$ and $S = s$, we have*

$$\|H - H_{\theta^0, \theta^0}\|_{S,0} \leq c_h m^{-\frac{1}{2} \frac{\alpha}{1+\alpha}} = h^\alpha. \tag{32}$$

6. *There is a bound $\Delta_{sample}(m, N)$ and sufficiently large constant $c_A$ so that*

$$\sup_{\substack{\|\theta^0 - \theta\|_* \leq c_A h \\ \|\theta^0 - \bar{\theta}\|_* \leq c_A h}} - \sum_{r=1}^{R} \langle \kappa^k, \partial_r f_{\bar{\theta}} \rangle_S [\partial_r \ell(\theta) - \partial_r \ell_0(\theta)] \leq \Delta_{sample}(m, N) \|\kappa^k\|_0 \|\kappa^k\|_S \tag{33}$$

*for all gradient descent iterates $\kappa^k$ with $k \leq n$.*

*Then, while the gradient descent error exceeds the final approximation and estimation error*

$$\|\kappa^k\|_0^2 \geq c_a \left( m^{-\frac{1}{2} \frac{\alpha}{1+\alpha}} + \Delta_{sample}(m, N) \right)^{\frac{s}{\beta}} \|\kappa^0\|_s^2, \qquad k \leq n, \tag{34}$$

*we have*

$$\|\kappa^{n+1}\|_0^2 \leq C e^{-\gamma [h^\alpha + \Delta_{sample}(m, N)](n+1)} \|\kappa^0\|_0^2, \qquad \|\kappa^{n+1}\|_s^2 \leq C \|\kappa^0\|_s^2$$

*for sufficiently large constants $c_a$, $c$ and $C$ independent of $m$, $\kappa^0$ and $\kappa^{n+1}$.*

The theorem has a long list of assumptions, which we verify for the proof of the main theorems in Sections B and C below. The coercivity (28) is the only major assumption that remains in the main theorems. The second assumption (29) is used to show that the weights do not move far from their initial and the third (30) provides bounds for the learning rate. The next two assumptions (31) and (32) are major components of NTK analysis and require that the NTK is Hölder continuous and that the empirical NTK concentrates close to the infinite width limit. Finally, assumption (33) bounds the difference between the gradient of the continuous $L_2$ loss and the discrete loss $\ell$. The last assumption is the major concern of this paper, while all others have been established in Welper (2024b;a).

The proof is identical to Welper (2024a, Theorem 3.1), with one difference: The error reduction lemma Welper (2024a, Lemma 3.2) is replaced with Lemma A.2 below. This introduces a new error term form the sample loss. While this extra error term only requires minimal changes in the proof, for the convenience of the reader, we include it in Section A.3.

## A.2 Gradient Descent Error Reduction

The first step in our convergence proof establishes an error decay in every gradient descent step. It matches Welper (2024a, Lemma 3.2) up to an additional error term $\Delta_{sample}(m, J)$ for the difference between the continuous and discrete losses.

**Lemma A.2.** *Assume that* (30)*,* (31)*,* (32) *and* (33) *hold. Assume that $\|\theta^0 - \theta^n\|_* \leq h$. Then*

$$\ell_S(\theta^{n+1}) - \ell_S(\theta^n) \leq -\gamma \langle \kappa, H\kappa \rangle_S + c\gamma h^\alpha \|\kappa\|_0 \|\kappa\|_S + \gamma \Delta_{sample}(m, N) \|\kappa\|_0 \|\kappa\|_S.$$

*Proof.* Applying the mean value theorem to the gradient descent step $\theta^{n+1} = \theta^n - \Delta^n$ with $\Delta^n := \gamma \nabla_\theta \ell(\theta^n)$, we obtain

$$\ell_S(\theta^{n+1}) - \ell_S(\theta^n) = \ell_S(\theta^n - \Delta^n) - \ell_S(\theta^n)$$
$$= -\ell'_S(\theta^n - \xi \Delta^n) \Delta^n,$$

for some $\xi \in (0, 1)$. Abbreviating $\kappa = \kappa^n$ and plugging in the derivatives $\ell'_S(\theta)^T = [\langle \kappa, \partial_r f_\theta \rangle_S]_{r=1}^{R}$ and $\Delta^n = \gamma [\partial_r \ell(\theta)]_{r=1}^{R}$, yields

$$\ell_S(\theta^{n+1}) - \ell_S(\theta^n) = -\gamma \sum_r \langle \kappa, \partial_r f_{\theta^n - \xi \Delta^n} \rangle_S \partial_r \ell(\theta^n).$$

Next, we replace the derivative $\partial_r \ell(\theta)$ of the discrete loss by the corresponding derivative of the continuous loss $\partial_r \ell_0(\theta) = \langle \kappa, \partial_r f_\theta \rangle$

$$
\begin{aligned}
\ell_S(\theta^{n+1}) - \ell_S(\theta^n) &= -\gamma \sum_r \langle \kappa, \partial_r f_{\theta^n - \xi \Delta^n} \rangle_S \left[ \partial_r \ell_0(\theta^n) + [\partial_r \ell(\theta^n) - \partial_r \ell_0(\theta^n)]] \right. \\
&= -\gamma \sum_r \langle \kappa, \partial_r f_{\theta^n - \xi \Delta^n} \rangle_S \langle \kappa, \partial_r f_{\theta^n} \rangle \\
&\quad - \gamma \sum_r \langle \kappa, \partial_r f_{\theta^n - \xi \Delta^n} \rangle_S [\partial_r \ell(\theta^n) - \partial_r \ell_0(\theta^n)] \\
&=: (I) + (II).
\end{aligned}
$$

Let us first estimate $(I)$. To this end, we express $\langle v, \kappa \rangle$ by the $\mathcal{H}^0$ dual $v^* \kappa := \langle v, \kappa \rangle$ and obtain

$$
\begin{aligned}
(I) &= -\gamma \sum_r \langle \kappa, \partial_r f_{\theta^n - \xi \Delta^n} \rangle_S \, \partial_r (f_{\theta^n})^*(\kappa) \\
&= -\gamma \left\langle \kappa, \left[ \sum_r (\partial_r f_{\theta^n - \xi \Delta^n})(\partial_r f_{\theta^n})^* \right] \kappa \right\rangle_S, \\
&= -\gamma \langle \kappa, H_{\theta^n - \xi \Delta^n, \theta^n} \kappa \rangle_S.
\end{aligned}
$$

Adding and subtracting terms to compare $f_{\theta^n - \xi \Delta^n}$ and $f_{\theta^n}$ with the initial $f_{\theta^0}$, we obtain

$$
\begin{aligned}
(I) &= -\gamma \langle \kappa, H_{\theta^0, \theta^0} \kappa \rangle_S \\
&\quad + \gamma \langle \kappa, H_{\theta^0, \theta^0} - H_{\theta^0, \theta^n} \kappa \rangle_S \\
&\quad + \gamma \langle \kappa, H_{\theta^0, \theta^n} - H_{\theta^n - \xi \Delta^n, \theta^n} \kappa \rangle_S.
\end{aligned}
$$

Assumption (32) implies

$$
- \langle \kappa, H_{\theta^0, \theta^0} \kappa \rangle_S = - \langle \kappa, H \kappa \rangle_S + \langle \kappa, H - H_{\theta^0, \theta^0} \kappa \rangle_S \leq - \langle \kappa, H \kappa \rangle_S + h^\alpha \|\kappa\|_0 \|\kappa\|_S
$$

and Assumption (31), with $\bar{h} = h$ and $\bar{h} = h + \|\Delta^n\|_*$ implies

$$
\begin{aligned}
\|H_{\theta^0, \theta^0} - H_{\theta^0, \theta^n}\|_{S,0} &\leq c h^\alpha, \\
\|H_{\theta^0, \theta^n} - H_{\theta^n - \xi \Delta^n, \theta^n}\|_{S,0} &\leq c \left( h + \|\Delta^n\|_* \right)^\alpha.
\end{aligned}
$$

Combining these inequalities, yields

$$
\begin{aligned}
(I) &\leq -\gamma \langle \kappa, H \kappa \rangle_S + 3c\gamma \left[ h + \|\Delta^n\|_* \right]^\alpha \|\kappa\|_0 \|\kappa\|_S, \\
&\leq -\gamma \langle \kappa, H \kappa \rangle_S + c\gamma h^\alpha \|\kappa\|_0 \|\kappa\|_S,
\end{aligned}
$$

where in the last step we have used that $\|\Delta^n\|_* = \gamma \|\nabla_\theta \ell(\theta^n)\|_* \lesssim h$ by assumption (30).

It remains to bound $(II)$. Again, using $\|\Delta^n\|_* \lesssim h$ and the assumptions of this lemma, we have $\|\theta^0 - (\theta^n + \xi \Delta^n)\| \lesssim h$ for all $\xi \in [0, 1]$. Thus, with Assumption (33) and our abbreviation $\kappa = \kappa^n$, we obtain

$$
\begin{aligned}
(II) &= -\gamma \sum_r \langle \kappa, \partial_r f_{\theta^n - \xi \Delta^n} \rangle_S [\partial_r \ell(\theta^n) - \partial_r \ell_0(\theta^n)] \\
&\leq \gamma \Delta_{sample}(m, N) \|\kappa\|_0 \|\kappa\|_S.
\end{aligned}
$$

Combining $(I)$ and $(II)$ yields the lemma.

$\square$

## A.3 Proof of Theorem A.1

The following proof is identical to Welper (2024a, Theorem 3.1) up to the additional error term $\Delta_{sample}(m, N)$. We include the proof to trace the minor modification and keep the paper self contained.

*Proof of Theorem A.1.* The proof is based on the gradient descent error reduction in Lemma A.2. All of its assumptions are given, except the weight distance $\left\| \theta^n - \theta^0 \right\|_*$, which we include in the induction hypothesis: We assume

$$\|\kappa^k\|_0^2 \lesssim e^{-\gamma[h^\alpha + \Delta]k}\|\kappa^k\|_0^2,$$

$$h^k := \max_{l \leq k}\left\| \theta^l - \theta^0 \right\|_* \lesssim c_h m^{-\frac{1}{2}\frac{1}{1+\alpha}} =: h$$

for all $k \leq n$ and $\Delta = \Delta_{sample}(m, N)$ and prove the case $k = n + 1$ by induction. With the induction hypothesis and assumptions (30), (31), (32) and (33), Lemma A.2 together with coercivity (1) implies

$$\|\kappa^{n+1}\|_0^2 - \|\kappa^0\|_0^2 \leq -\gamma\|\kappa^n\|_{-\beta}^2 + c\gamma h^\alpha\|\kappa^n\|_0^2 + \gamma\Delta\|\kappa^n\|_0^2.$$

$$\|\kappa^{n+1}\|_s^2 - \|\kappa^0\|_s^2 \leq -\gamma\|\kappa^n\|_{s-\beta}^2 + c\gamma h^\alpha\|\kappa^n\|_0\|\kappa^n\|_s + \gamma\Delta\|\kappa^n\|_0\|\kappa^n\|_s,$$

or shorter

$$\|\kappa^{n+1}\|_0^2 - \|\kappa^0\|_0^2 \leq -\gamma\|\kappa^n\|_{-\beta}^2 + \gamma[ch^\alpha + \Delta]\|\kappa^n\|_0^2.$$

$$\|\kappa^{n+1}\|_s^2 - \|\kappa^0\|_s^2 \leq -\gamma\|\kappa^n\|_{s-\beta}^2 + \gamma[ch^\alpha + \Delta]\|\kappa^n\|_0\|\kappa^n\|_s.$$

This is not a closed iteration of the $\|\kappa^n\|_0^2$ and $\|\kappa\|_s^2$ residuals because of the $\|\cdot\|_{-\beta}$ and $\|\cdot\|_{s-\beta}$ norms. We eliminate them with the interpolation inequalities 27

$$\|\kappa\|_0 \leq \|\kappa\|_{-\beta}^{\frac{s}{\beta+s}}\|\kappa\|_s^{\frac{\beta}{\beta+s}} \qquad \Rightarrow \qquad -\|\kappa\|_{-\beta}^2 \leq -\|\kappa\|_0^{2+\frac{2\beta}{s}}\|\kappa\|_s^{-\frac{2\beta}{s}},$$

$$\|\kappa\|_0 \leq \|\kappa\|_{s-\beta}^{\frac{s}{\beta}}\|\kappa\|_s^{\frac{\beta-s}{\beta}} \qquad \Rightarrow \qquad -\|\kappa\|_{s-\beta}^2 \leq -\|\kappa\|_0^{\frac{2\beta}{s}}\|\kappa\|_s^{2-\frac{2\beta}{s}}$$

so that

$$\|\kappa^{n+1}\|_0^2 - \|\kappa^0\|_0^2 \lesssim -\gamma\|\kappa^n\|_0^{2+\frac{2\beta}{s}}\|\kappa^n\|_s^{-\frac{2\beta}{s}} + \gamma[h^\alpha + \Delta]\|\kappa^n\|_0^2,$$

$$\|\kappa^{n+1}\|_s^2 - \|\kappa^0\|_s^2 \lesssim -\gamma\|\kappa^n\|_0^{\frac{2\beta}{s}}\|\kappa^n\|_s^{2-\frac{2\beta}{s}} + \gamma[h^\alpha + \Delta]\|\kappa^n\|_0\|\kappa^n\|_s.$$

Error bounds for this iteration are given in Lemma D.5 with $n + 1 := \|\kappa^{n+1}\|_0^2$, $v^{n+1} := \|\kappa\|_s^2$, $\rho = \beta/s$, $a = c = 1$ and $b = d = h^\alpha + \Delta$. To show the lemma's assumption (50), we use $2s \leq \beta$ so that

$$\left(2 - \frac{s}{\beta}\right) \leq 2 \qquad \Leftrightarrow \qquad \frac{s}{\beta} \leq \frac{2\frac{s}{\beta}}{2 - \frac{s}{\beta}} \qquad \Leftrightarrow \qquad \frac{1}{\rho} \leq \frac{2}{2\rho - 1}.$$

Hence, assumption (34) implies

$$u^k = \|\kappa^k\|_0^2 \geq \left(\left(m^{-\frac{1}{2}\frac{1}{1+\alpha}}\right)^\alpha + \Delta\right)^{\frac{s}{\beta}}\|\kappa^0\|_s^2 = (h^\alpha + \Delta)^{\frac{s}{\beta}}\|\kappa^0\|_s^2 \gtrsim \left(2\frac{b}{a}\right)^{\frac{1}{\rho}}v^0 \gtrsim \left(\frac{d}{c}\right)^{\frac{2}{2\rho-1}}v^0.$$

and Lemma D.5 is applicable. Therefore, we obtain

$$\|\kappa^{n+1}\|_0^2 \lesssim e^{-\gamma[h^\alpha + \Delta](n+1)}\|\kappa^0\|_0^2, \qquad\qquad \|\kappa^{n+1}\|_s^2 \lesssim \|\kappa^0\|_s^2,$$

which shows the first induction hypothesis.

It remains to bound $h^{n+1}$ to show the second induction hypothesis. To this end note that

$$h^{n+1} = \max_{k \leq n+1}\left\| \theta^k - \theta^0 \right\|_* \lesssim \frac{\gamma}{\sqrt{m}}\sum_{k=1}^n\|\kappa^k\|_0 \lesssim \frac{\gamma}{\sqrt{m}}\sum_{k=1}^n e^{-\gamma[h^\alpha + \Delta]k}\|\kappa^0\|_0,$$

where in the second step we have used assumption (29) and in the third step the induction hypothesis. We bound the latter sum

$$\sum_{k=1}^n e^{-\gamma[h^\alpha + \Delta]k} \leq \int_0^\infty e^{-\gamma[h^\alpha + \Delta]k}\,dk = \frac{1}{\gamma[h^\alpha + \Delta]},$$

to conclude that

$$h^{n+1} \leq c \frac{\gamma}{\sqrt{m}} \frac{1}{\gamma[h^\alpha + \Delta]} \|\kappa^0\|_0 \leq c \frac{\gamma}{\sqrt{m}} \frac{1}{\gamma h^\alpha} \|\kappa^0\|_0 \leq \frac{c}{\sqrt{m}} h^{-\alpha} \|\kappa^0\|_0 = h.$$

In the last step we have used that the definition of $h$ implies

$$h = c_h m^{-\frac{1}{2}\frac{1}{1+\alpha}} = \frac{c_h}{\sqrt{m}} m^{\frac{1}{2}\frac{\alpha}{1+\alpha}} = \frac{c}{\sqrt{m}} m^{\frac{1}{2}\frac{\alpha}{1+\alpha}} \|\kappa^0\|_0 = \frac{c}{\sqrt{m}} h^{-\alpha} \|\kappa^0\|_0$$

for constant $c_h = c\|\kappa^0\|_0$ dependent on $\|\kappa^0\|_0$. Together with the first induction hypothesis this concludes the proof.

$\square$

## B    Sampling

In this section, we prove bounds for the error $\Delta_{sample}(m, N)$ between continuous and sample loss (Lemma B.4) and then Theorem 2.2. Throughout the section, we use the following regularity conditions on the activation function

$$|\sigma(x)| \lesssim |x|, \tag{35}$$

$$|\sigma(x) - \sigma(\bar{x})| \lesssim |x - \bar{x}| \tag{36}$$

$$|\dot{\sigma}(x)| \lesssim 1. \tag{37}$$

Moreover, for the time being, we assume that the weight matrices are bounded

$$\|W^\ell\| n_\ell^{-1/2} \lesssim 1, \qquad \|\bar{W}^\ell\| n_\ell^{-1/2} \lesssim 1, \qquad \|x\| \lesssim 1 \, \forall x \in D. \tag{38}$$

Typically $W^\ell$ will be the initial weight and $\bar{W}^\ell$ the weight of a later gradient descent step. Likewise, we denote by $\bar{\theta}$, $f_{\bar{\theta}}$ and $\bar{f}^\ell$, etc. the weights, networks and layers based on the perturbed $\bar{W}^\ell$. For the main theorems, these bounds follow from properties of random matrices at the initial weights and the observation that weights do not move far from their initial in (29).

Throughout the section, we abbreviate $D = \mathbb{S}^{d-1}$, $L_2 = L_2(D)$, $H^s = H^s(D)$ and $\langle \cdot, \cdot \rangle_{H^s} = \langle \cdot, \cdot \rangle_{H^s(D)}$, when convenient.

### B.1    Concentration

In this section, for the sample loss $\ell(\theta) = \frac{1}{2N} \sum_{i=1}^N |f_\theta(x_i) - f(x_i)|^2$, we bound the sample error

$$\sup_{\theta, \bar{\theta}} \left| \sum_{r=1}^R \left\langle \kappa^k, \partial_r f_{\bar{\theta}} \right\rangle_{H^s} [\partial_r \ell(\theta) - \partial_r \ell_0(\theta)] \right|.$$

This establishes assumption (33) in the abstract convergence Theorem A.1 and leads to the proof of the first main result.

Let us abbreviate

$$Y_\theta := \sum_{r=1}^R \left\langle \kappa, \partial_r f_{\bar{\theta}} \right\rangle_{H^s} [\partial_r \ell(\theta) - \partial_r \ell_0(\theta)],$$

which is a random variable with respect to the sample points $x_i$ implicitly contained in the discrete loss $\ell(\theta)$. Since the $\ell_0$ loss is the expectation of the sample loss $\ell$, it is easy to see that $\mathbb{E}[Y_\theta] = 0$ and is suffices to show concentration results. These follow from Dudley's inequality, for which we proof that $Y_\theta$ has sub-gaussian increments, i.e.

$$\|Y_\theta - Y_\vartheta\|_{\psi_2} \lesssim \|\theta - \vartheta\|_*.$$

Throughout the section, we use Orlicz and weight norms

$$\|X\|_{\psi_2} = \inf\left\{t > 0 : \mathbb{E}\left[\exp(X^2/t^2)\right] \le 2\right\},$$
$$\|\theta\|_* := \|W^{L-1}\|m_{L-1}^{-1/2},$$

where we have used that $\theta = W^{L-1}$ because we only train the second but last layer. $\|\cdot\|$ denotes the Euclidean norm for vectors and the induced matrix norm for matrices. As in the introduction, we abbreviate $D := \mathbb{S}^{d-1}$.

Before we prove sub-gaussian increments, we bound several components involved in $Y_\theta$ separately. We start with the factor $\langle \kappa, \partial_r f_\theta \rangle_{H^s}$.

**Lemma B.1.** *Let $0 \le s < 1$ and assume $\sigma$ satisfies (35), (36), the weights satisfy (38) and are of equivalent size $m_0 \sim \cdots \sim m_{L-1}$. Then*

$$\sum_{r=1}^{R} \langle \kappa, \partial_r f_\theta \rangle_{H^s}^2 \lesssim \|\kappa\|_{H^s}^2.$$

*Proof.* By Lemma D.2 cited from Welper (2024b), with $\epsilon$ sufficiently small so that $s + \epsilon < 1$ the bilinear form

$$B(u,v) = \int_D \int_D u(x) \left( \sum_{r=1}^{R} \partial_r f_\theta(x) \partial_r f_\theta(y) \right) v(y)\, dx dy$$

$$= \sum_{r=1}^{R} \langle u, \partial_r f_\theta \rangle \langle \partial_r f_\theta, v \rangle$$

for $u$ and $v$ in $L_2$ is bounded by

$$B(u,v) \lesssim \|u\|_{H^{-s}} \|v\|_{H^{-s}}$$

and can therefore be extended to all $u, v \in H^{-s}$. In this case the $L_2$ inner product $\langle \cdot, \cdot \rangle$ turns into the $H^{-s} \times H^s$ dual pairing. Denoting by $R\colon H^s \to H^{-s}$ the Riesz map, we obtain

$$\sum_{r=1}^{R} \langle \kappa, \partial_r f_\theta \rangle_{H^s}^2 = \sum_{r=1}^{R} \langle \kappa, \partial_r f_\theta \rangle_{H^s} \langle \partial_r f_\theta, \kappa \rangle_{H^s}$$

$$= \sum_{r=1}^{R} \langle R\kappa, \partial_r f_\theta \rangle \langle \partial_r f_\theta, R\kappa \rangle = B(R\kappa, R\kappa) \lesssim \|R\kappa\|_{H^{-s}}^2 = \|\kappa\|_{H^s}^2,$$

which proves the lemma

$\square$

Next, we consider the loss $\partial_r \ell(\theta) = \frac{1}{N} \sum_{i=i}^{N} \kappa(x_i) \partial_r f_\theta(x_i)$ and show sub-gaussian increments for the summands. As usual, we abbreviate $L_\infty(D) = L_\infty$.

**Lemma B.2.** *Let $0 < s < 1$ and assume $\|f\|_{L_\infty} \lesssim m^{1/2}$, $\sigma$ satisfies (35), (36), (37) and the weights satisfy (38). Let $X$ be a uniform random variable with values in $D$ and set*

$$X_\theta := \kappa(X) \nabla f_\theta(X) - \mathbb{E}\left[\kappa(X) \nabla f_\theta(X)\right]$$

*Then*

$$\left\| \|X_\theta - X_\vartheta\| \right\|_{\psi_2} \lesssim m^{1/2} \|\theta - \vartheta\|_*, \qquad\qquad \left\| \|X_\theta\| \right\|_{\psi_2} \lesssim m^{1/2}.$$

Note that $\kappa(X)$ is a scalar and $\nabla f_\theta(X) = \nabla_{W^{L-1}} f_\theta(X)$ is a matrix. With the convention $\partial_{W_{ij}^{L-1}} = \partial_r$ for suitable $r$, we may also regard $\nabla f_\theta(X)$ as a vector and thus $\|\kappa(X) \nabla f_\theta(X)\|$ as the Euclidean vector norm, which we do in the following.

*Proof.* We split $X_\theta = Z_\theta - \mathbb{E}[Z_\theta]$ into a random part and an expectation, with

$$Z_\theta := \kappa(X) \nabla f_\theta(X)$$

and estimate both terms separately.

1. *Estimate $\big\| \|Z_\theta - Z_\vartheta\| \big\|_{\psi_2}$:* First, we upper bound the $\psi_2$-norm by the $L_\infty$-norm and separate the factor $\kappa$:

$$\big\| \|Z_\theta - Z_\vartheta\| \big\|_{\psi_2}^2 \leq \big\| \|Z_\theta - Z_\vartheta\| \big\|_{L_\infty}^2 = \big\| \|\kappa \nabla f_\theta - \kappa \nabla f_\vartheta\| \big\|_{L_\infty}^2$$

$$= \sup_{x \in D} \sum_{r=1}^{R} \left[ \kappa(x) \left[ \partial_r f_\theta(x) - \partial_r f_\vartheta(x) \right] \right]^2 \tag{39}$$

$$\leq \|\kappa\|_{L_\infty}^2 \big\| \|\nabla f_\theta - \nabla f_\vartheta\| \big\|_{L_\infty}^2 .$$

The first factor is bounded by

$$\|\kappa\|_{L_\infty} \leq \|f\|_{L_\infty} + \|f_{\bar\theta}\|_{L_\infty} \lesssim m^{1/2},$$

where we have used that $\kappa = f - f_{\bar\theta}$ for some weights $\bar\theta$ that satisfy (38), that $\|f\|_{L_\infty} \lesssim m^{-1/2}$ by assumption and $\|f_{\bar\theta}\|_{L_\infty} \lesssim m^{-1/2}$ by Lemma D.3, cited from Welper (2024b).

To bound the second factor, recall that the derivatives $\partial_r$ for $r \in \{1, \dots, R\}$ are shorthand for the derivatives $\partial_{W_{ij}^{L-1}}$ of the second but last layer. A short calculation shows that

$$\partial_{W_{ij}^{L-1}} f_\theta = \underbrace{W_i^L m_L^{-1/2} m_{L-1}^{-1/2}}_{:=w_i} \underbrace{\dot\sigma\left(f_i^L\right)}_{=:u_i(\theta)} \underbrace{\sigma\left(f_j^{L-1}\right)}_{=:v_j}, \tag{40}$$

see e.g. Welper (2024b, Proof of Lemma 6.18) for details. The weights $W^L$ of the last layer have only one index because the network is scalar valued. The layer $f^{L-1}$ does not depend on $W^{L-1}$ and therefore $v_j$ does not depend on $\theta = W^{L-1}$. Then, for all $x \in D$, we have

$$\|\nabla f_\theta(x) - \nabla f_\vartheta(x)\|^2 = \sum_{ij} \left[ w_i u_i(\theta) v_j - w_i u_i(\vartheta) v_j \right]^2$$

$$= \sum_{ij} w_i^2 \left[ u_i(\theta) - u_i(\vartheta) \right]^2 v_j^2$$

$$= \left[ \sum_i w_i^2 \left[ u_i(\theta) - u_i(\vartheta) \right]^2 \right] \left[ \sum_j v_j^2 \right]$$

$$\leq \|w\|_{\ell_\infty}^2 \|u(\theta) - u(\vartheta)\|^2 \|v\|^2 .$$

Since $W_i^L = \pm 1$, we have

$$\|w\|_{\ell_\infty} \lesssim m^{-1}$$

and from Lemma D.3 cited from Welper (2024b), together with the Lipschitz continuity of $\sigma$ and $\dot\sigma$, we have

$$\|u(\theta) - u(\vartheta)\| \lesssim m^{1/2} \|\theta - \vartheta\|_* , \qquad\qquad \|v\| \lesssim m^{1/2} .$$

Thus, we obtain

$$\|\nabla f_\theta(x) - \nabla f_\vartheta(x)\| \lesssim \|\theta - \vartheta\|_*$$

for all $x \in D$ and therefore together with (39) that

$$\big\| \|Z_\theta - Z_\vartheta\| \big\|_{\psi_2} \lesssim m^{1/2} \|\theta - \vartheta\|_* .$$

2. *Estimate* $\|\mathbb{E}\left[Z_\theta\right] - \mathbb{E}\left[Z_\vartheta\right]\|$: First, using that $X$ is uniform, we factor out the residual $\kappa$:

$$\|\mathbb{E}\left[Z_\theta\right] - \mathbb{E}\left[Z_\vartheta\right]\|^2 \leq \sum_{r=1}^{R} \left[\frac{1}{|D|} \int_D \kappa(x)\left[\partial_r f_\theta(x) - \partial_r f_\vartheta(x)\right] dx\right]^2$$

$$= \frac{1}{|D|}\|\kappa\|_{L_2}^2 \sum_{r=1}^{R} \frac{1}{|D|} \int_D |\partial_r f_\theta(x) - \partial_r f_\vartheta(x)|^2 \, dx$$

$$\leq \|\kappa\|_{L_\infty}^2 \big\| \|\partial_r f_\theta - \partial_r f_\vartheta\| \big\|_{L_\infty}^2$$

where in the first inequality we have used Cauchy-Schwarz and in the last one exchanged the order of sum and integral. The right hand side is identical to (39) and bounded the same way.

Combining the estimates for $Z_\theta$ and its expectation and using that the $\psi_2$-norm of a non-negative constant is upper bounded by the constant itself, we obtain

$$\big\| \|X_\theta - X_\vartheta\| \big\|_{\psi_2} \leq \big\| \|Z_\theta - Z_\vartheta\| \big\|_{\psi_2} + \big\| \|\mathbb{E}\left[Z_\theta\right] - \mathbb{E}\left[Z_\vartheta\right]\| \big\|_{\psi_2} \lesssim m^{1/2} \|\theta - \vartheta\|_* ,$$

which shows the first part of the lemma.

The proof of the second bound $\big\| \|X_\theta\| \big\|_{\psi_2} \lesssim m^{1/2}$ is identical upon replacing $X_\vartheta$ and $Z_\vartheta$ with $0$ throughout the proof. Using $\dot{\sigma}(x) \lesssim 1$, we obtain $\|u(\theta)\| \lesssim m^{1/2}$ instead of $\|u(\theta) - u(\vartheta)\| \lesssim m^{1/2} \|\theta - \vartheta\|_*$ by Lemma D.3. Omitting the factor $\|\theta - \vartheta\|_*$ in the left hand side throughout the rest of the proof shows the second part of the lemma.

$\square$

The next lemma establishes the sub-gaussian increments of the random variable $Y_\theta$.

**Lemma B.3.** *Let* $0 < s < 1$ *and assume* $\|f\|_{L_\infty} \lesssim m^{1/2}$, $\sigma$ *satisfies* (35), (36), (37) *and the weights satisfy* (38). *Let* $X$ *be a uniform random variable on* $D$ *and for* $\theta, \bar{\theta} \in \Theta$ *set*

$$Y_\theta := \sum_{r=1}^{R} \langle \kappa, \partial_r f_{\bar{\theta}} \rangle_{H^S} \left[\partial_r \ell(\theta) - \partial_r \ell_0(\theta)\right].$$

*Then*

$$\|Y_\theta - Y_\vartheta\|_{\psi_2} \lesssim \left(\frac{m}{N}\right)^{1/2} \|\kappa\|_{H^S} \|\theta - \vartheta\|_* , \qquad \|Y_\theta\|_{\psi_2} \lesssim \left(\frac{m}{N}\right)^{1/2} \|\kappa\|_{H^S}.$$

*Proof.* Plugging in the definitions of the loss $\ell$ and its continuum limit $\ell_0$, we have

$$\partial_r \ell(\theta) = \frac{1}{N} \sum_{i=1}^{N} \kappa(x_i)\partial_r f_\theta(x_i), \qquad \partial_r \ell_0(\theta) = \langle \kappa, \partial_r f_\theta \rangle = \mathbb{E}\left[\kappa \partial_r f_\theta\right]$$

and therefore

$$Y_\theta = \frac{1}{N} \sum_{i=1}^{N} \sum_{r=1}^{R} \underbrace{\langle \kappa, \partial_r f_{\bar{\theta}} \rangle_{H^S}}_{=:u_r} \underbrace{\left[\kappa(x_i)\partial_r f_\theta(x_i) - \mathbb{E}\left[\kappa(X)\partial_r f_\theta(X)\right]\right]}_{=:(X_\theta^i)_r} = \frac{1}{N} \sum_{i=1}^{N} u^T X_\theta^i.$$

With Hoeffding's inequality, we estimate the $\psi_2$-norm by

$$\|Y_\theta - Y_\vartheta\|_{\psi_2}^2 = \left\| \frac{1}{N} \sum_{i=1}^{N} u^T (X_\theta^i - X_\vartheta^i) \right\|_{\psi_2}^2 \lesssim \frac{1}{N^2} \sum_{i=1}^{N} \left\| u^T (X_\theta^i - X_\vartheta^i) \right\|_{\psi_2}^2.$$

The argument of the $\psi_2$-norm is bounded by $|u^T(X_\theta^i - X_\vartheta^i)| \le \|u\| \|X_\theta^i - X_\vartheta^i\|$ and thus

$$\left\| Y_\theta - Y_\vartheta \right\|_{\psi_2}^2 \le \frac{1}{N^2} \sum_{i=1}^{N} \|u\|^2 \left\| \|X_\theta^i - X_\vartheta^i\| \right\|_{\psi_2}^2$$

$$\lesssim \frac{1}{N^2} \sum_{i=1}^{N} \|\kappa\|_{H^s}^2 m \|\theta - \vartheta\|_*^2$$

$$\lesssim \frac{m}{N} \|\kappa\|_{H^s}^2 \|\theta - \vartheta\|_*^2 ,$$

where the last inequalities follows from

$$\|u\| \lesssim \|\kappa\|_{H^s} , \qquad\qquad \left\| \|X_\theta^i - X_\vartheta^i\| \right\|_{\psi_2} \lesssim m^{1/2} \|\theta - \vartheta\|_* ,$$

by Lemmas B.1 and B.2, respectively. Taking the square root completes the proof of the first inequality. The second follows analogously by replacing $Y_\vartheta$ and $X_\vartheta$ with zero throughout the proof.

$\square$

**Lemma B.4.** *Let $0 < s < 1$ and assume $\|f\|_{L_\infty} \lesssim m^{1/2}$, $\sigma$ satisfies (35), (36), (37) and the weights satisfy (38). Let $\Theta_h := \{\theta \in \Theta \mid \left\| \theta - \theta^0 \right\|_* \le h\}$ for $h \lesssim 1$ and some initial (trained) weight $\theta^0$ that satisfies (38). Then with probability at least $1 - 2\exp(-m^2 h^2)$*

$$\sup_{\theta \in \Theta_h} \left| \sum_{r=1}^{R} \langle \kappa, \partial_r f_{\bar\theta} \rangle_{H^s} \left[ \partial_r \ell(\theta) - \partial_r \ell_0(\theta) \right] \right| \lesssim \frac{m^{3/2}}{N^{1/2}} h \|\kappa\|_{H^s} .$$

*Proof.* We abbreviate

$$Y_\theta := \sum_{r=1}^{R} \langle \kappa, \partial_r f_{\bar\theta} \rangle_{H^s} \left[ \partial_r \ell(\theta) - \partial_r \ell_0(\theta) \right]$$

and prove the lemma with Dudley's inequality. We have established in Lemma B.3 that $Y_\theta$ has sub-gaussian increments

$$\left\| Y_\theta - Y_\vartheta \right\|_{\psi_2} \lesssim \left( \frac{m}{N} \right)^{1/2} \|\kappa\|_{H^s} \|\theta - \vartheta\|_* .$$

Next, we estimate the covering numbers in Dudley's inequality. The set of eligible parameters $\Theta_h$ is contained in the ball of radius $h$ in $\mathbb{R}^R$ in the $\|\cdot\|_*$-norm. Hence, for every $\epsilon \ge 0$ there is an $\epsilon$-covering of at most

$$N(\epsilon) \le \left( \frac{3h}{\epsilon} \right)^R$$

$\epsilon$-balls, see e.g. Lorentz et al. (1996, Chapter 15, Proposition 1.3). Then, using $\log x \le x - 1 \le x$,

$$\int_0^\infty \sqrt{\log N(\epsilon)} \, d\epsilon \;=\; R^{1/2} \int_0^h \sqrt{\log\left(\frac{3h}{\epsilon}\right)} \, d\epsilon \;\le\; R^{1/2} \int_0^h \sqrt{\left(\frac{3h}{\epsilon}\right)} \, d\epsilon \;=\; \frac{1}{2} (3hR)^{1/2} h^{1/2} \;\lesssim\; mh,$$

where in the last step we have used that $R^{1/2} \sim m$. Hence, Dudley's inequality (with tail bounds) implies that for all $u \ge 0$

$$\sup_{\theta, \vartheta \in \Theta_h} |Y_\theta - Y_\vartheta| \lesssim \left( \frac{m}{N} \right)^{1/2} \|\kappa\|_{H^s} \left[ \int_0^\infty \sqrt{\log N(\epsilon)} \, d\epsilon + uh \right]$$

holds with probability at least $1 - 2\exp(-u^2)$. Choosing $u = m$ yields

$$\sup_{\theta, \vartheta \in \Theta_h} |Y_\theta - Y_\vartheta| \lesssim \left( \frac{m}{N} \right)^{1/2} mh \|\kappa\|_{H^s} ,$$

with probability at least $1 - 2\exp(-m^2)$. The sub-gaussian bound $\left\|Y_\theta\right\|_{\psi_2} \lesssim \left(\frac{m}{N}\right)^{1/2}\|\kappa\|_{H^s}$ from Lemma B.3 implies

$$|Y_{\theta^0}| \lesssim \left(\frac{m}{N}\right)^{1/2} mh\|\kappa\|_{H^s}.$$

with probability at least $1 - 2\exp(-m^2h^2)$. Then, the lemma follows from $\sup_{\theta\in\Theta_h}|Y_\theta| \leq \sup_{\theta\in\Theta_h}|Y_\theta - Y_{\theta^0}| + |Y_{\theta^0}|$.

$\square$

## B.2 Convergence: Proof of Theorem 2.2

*Proof of Theorem 2.2.* The theorem follows from Theorem A.1 with $\mathcal{H}^s = H^s$ and $\mathcal{H}^0 = L_2$, for which we have to verify all assumptions. Most of them have been established in the proof of Welper (2024a, Theorem 2.2), which is identical to the one of this paper, except that it uses a continuous $L_2$ loss instead of a sample loss. This results in the extra assumption (33), which is the only one left to verify.

By Lemma B.4, with probability at least $1 - 2\exp(-m^2h^2)$ we have

$$\sup_{\theta\in\Theta_h}\left|\sum_{r=1}^R \left\langle\kappa^k, \partial_r f_{\bar\theta}\right\rangle_{H^s}[\partial_r\ell(\theta) - \partial_r\ell_0(\theta)]\right| \lesssim \frac{m^{3/2}}{N^{1/2}}h\|\kappa^k\|_{H^s},$$

which provides an estimate for $\Delta_{sample}(m, N)$ up to a missing factor $\|\kappa^k\|_{L_2}$. To insert this factor, we use the lower bound from Assumption (34): For $k \leq n$

$$\|\kappa^k\|_{L_2}^2 \geq c_a\left(h^\alpha + \Delta_{sample}(m, N)\right)^{\frac{s}{\beta}}\|\kappa^0\|_{H^s}^2 \geq c_a h^{\alpha\frac{s}{\beta}}\|\kappa^0\|_{H^s}^2,$$

which implies

$$1 \lesssim h^{-\frac{\alpha s}{2\beta}}\|\kappa^0\|_{H^s}^{-1}\|\kappa^k\|_{L_2}.$$

Inserting this into the application of Lemma B.4 above, we obtain

$$\sup_{\theta\in\Theta_h}\left|\sum_{r=1}^R \left\langle\kappa^k, \partial_r f_{\bar\theta}\right\rangle_{H^s}[\partial_r\ell(\theta) - \partial_r\ell_0(\theta)]\right| \lesssim \left[\frac{m^{3/2}}{N^{1/2}}h^{1-\frac{\alpha s}{2\beta}}\|\kappa^0\|_{H^s}^{-1}\right]\|\kappa^k\|_{L_2}\|\kappa^k\|_{H^s}.$$

Comparing to the definition of $\Delta_{sample}(m, N)$ in assumption (33), we obtain

$$\Delta_{sample}(m, N) \lesssim \frac{m^{3/2}}{N^{1/2}}h^{1-\frac{\alpha s}{2\beta}}\|\kappa^0\|_{H^s}^{-1},$$

which shows the assumption.

It remains to consider the success probability. From the parts of the proof shown in Welper (2024a), the bounds fail with probability $cL(e^{-m} + e^{-\tau})$ and from above with probability $e^{-cm^2h^2} \leq e^{cm}$ because $h = c_h m^{-\frac{1}{2}\frac{1}{1+\alpha}} \geq m^{-1/2}$ for any $\alpha > 0$ from (30).

This completes the proof, together with an index shift $n + 1 \to n$ between the statements of Theorems A.1 and 2.2.

$\square$

## C Kernels

In this section, we prove Theorem 2.4 based on bounds for the error $\Delta_{sample}(m, N)$ between the $L_2(D)$ loss and the kernel loss

$$\ell^k(\theta) := \frac{1}{2N}\sum_{i=1}^N \langle k(x_i, \cdot), f_\theta - f\rangle^2$$

for kernels $k(x,y)$ with $x,y \in D = \mathbb{S}^{d-1}$ and $x_i$ uniformly and independently sampled on the domain $D$. We denote the corresponding expected loss by

$$\bar{\ell}^k(\theta) := \mathbb{E}\left[\ell^k(\theta)\right] := \frac{1}{2|D|} \int_D \langle k(x,\cdot), f_\theta - f \rangle^2 \, dx.$$

We consider this expectation in Section C.1, which is generally not identical to $\frac{1}{2}\|f_\theta - f\|^2_{L_2(D)}$. Then, we show corresponding concentration inequalities based on matrix Bernstein inequalities in Section C.2. The proof of Theorem 2.4 is in Section C.3.

Throughout this section, we abbreviate $D = \mathbb{S}^{d-1}$, $L_2 = L_2(D)$, $H^s = H^s(D)$ and $\langle \cdot, \cdot \rangle_{H^s} = \langle \cdot, \cdot \rangle_{H^s(D)}$, when convenient.

## C.1 Expectation

Define the inner product and norm that give rise to the expected kernel loss as

$$\langle u, v \rangle_k = \mathbb{E}\left[\frac{1}{N}\sum_{i=1}^N \langle u, k(x_i, \cdot)\rangle \langle k(x_i, \cdot), v\rangle\right], \qquad \|\cdot\|_k^2 := \langle \cdot, \cdot \rangle_k, \qquad \bar{\ell}^k(\theta) := \frac{1}{2}\|f_\theta - f\|_k^2. \tag{41}$$

Unfortunately, the norm $\|\cdot\|_k$ is not equivalent to the $L_2$ norm and therefore the expected loss $\bar{\ell}^k(\theta)$ is not equivalent to the $L_2$ loss $\ell_0(\theta)$. To bridge this gap, in this section, we construct a modified inner product and corresponding norm and loss

$$\langle u, v \rangle_\sharp \qquad\qquad \|\cdot\|_\sharp^2 := \langle \cdot, \cdot \rangle_\sharp, \qquad\qquad \bar{\ell}^\sharp(\theta) := \frac{1}{2}\|f_\theta - f\|_\sharp^2, \tag{42}$$

with the following properties:

1. The two inner products $\langle \cdot, \cdot \rangle_k$ and $\langle \cdot, \cdot \rangle_\sharp$ are close for smooth functions.

2. The norm $\|\cdot\|_\sharp^2$ and loss $\bar{\ell}^\sharp(\theta)$ are equivalent to $\|\cdot\|_{L_2}$ and $\ell_0(\theta)$, respectively.

To this end, we first characterize the inner product $\langle \cdot, \cdot \rangle_k$.

**Lemma C.1.** *Assume the symmetric kernel $k\colon D \times D \to \mathbb{R}$ has $L_2$-orthogonal eigenfunctions $\psi_j$ with eigenvalues $\lambda_j$. Then for any $u, v \in L_2$*

$$\langle u, v \rangle_k = \mathbb{E}\left[\frac{1}{N}\sum_{i=1}^N \langle u, k(\cdot, x_i)\rangle \langle k(x_i, \cdot), v\rangle\right] = \sum_{r=1}^\infty \lambda_j^2 \langle u, \psi_r \rangle \langle \psi_r, v \rangle.$$

*Proof.* We denote the expectation by $E$. By the law of large numbers, we have

$$E := \mathbb{E}\left[\frac{1}{N}\sum_{i=1}^N \langle u, k(\cdot, x_i)\rangle \langle k(x_i, \cdot), v\rangle\right] = \frac{1}{|D|}\int_D \langle u, k(\cdot, x)\rangle \langle k(x, \cdot), v\rangle \, dx.$$

Plugging in $u$, $v$ in eigenbasis

$$u = \sum_{s=1}^\infty \langle u, \psi_s \rangle \psi_s, \qquad\qquad v = \sum_{t=1}^\infty \langle u, \psi_t \rangle \psi_t,$$

we obtain

$$E = \sum_{s,t=1}^\infty \langle u, \psi_s \rangle \langle v, \psi_t \rangle \int_D \langle \psi_s, k(\cdot, x)\rangle \langle k(x, \cdot), \psi_t\rangle \, dx$$

$$= \sum_{s,t=1}^\infty \langle u, \psi_s \rangle \langle v, \psi_t \rangle \lambda_s \lambda_t \frac{1}{|D|} \int_D \psi_s(x)\psi_t(x) \, dx$$

$$= \sum_{s=1}^\infty \lambda_s^2 \langle u, \psi_s \rangle \langle \psi_s, v \rangle,$$

where in the second step we have used eigenvalue and vector definition $\langle \psi_s, k(\cdot, x) \rangle = \lambda_s \psi_s(x)$ and in the last step we have used that $\psi_s$ is a orthonormal basis, with normalized squared expectation.

$\square$

In the next lemma, we define the modified inner product $\langle \cdot, \cdot \rangle_\sharp$ and prove the properties from the introduction of this section.

**Lemma C.2.** *Assume the symmetric kernel $k \colon D \times D \to \mathbb{R}$ has $L_2$-orthogonal eigenfunctions $\psi_j$ and eigenvalues $\lambda_j$ with*

$$1 \lesssim \lambda_j \lesssim 1, \quad j \leq J,$$
$$\lambda_j \lesssim 1, \quad j > J.$$

*Define a lower bounded perturbation $\bar{\lambda}_j$ of the eigenvalue $\lambda_j$ and the corresponding inner product*

$$\bar{\lambda}_j := \left\{ \begin{array}{ll} \lambda_j & j \leq J \\ \max\{\lambda_j, 1\} & j \geq J. \end{array} \right. \qquad \langle u, v \rangle_\sharp := \sum_{r=1}^{\infty} \bar{\lambda}_j^2 \langle u, \psi_r \rangle \langle \psi_r, v \rangle .$$

*For some increasing weights $\mu_j \geq 1$ let*

$$\|v\|_{\bar{\mathcal{H}}^s}^2 := \sum_{j=1}^{J} \mu_j^{2s} \langle \psi_j, v \rangle^2$$

*be the norm of a Hilbert space $\bar{\mathcal{H}}^s$. Then for all $u, v \in L_2$ and $s \geq 0$*

$$\left| \langle u, v \rangle_k - \langle u, v \rangle_\sharp \right| = \left| \sum_{j=1}^{\infty} \left[ \lambda_j^2 - \bar{\lambda}_j^2 \right] \langle u, \psi_j \rangle \langle \psi_j, v \rangle \right| \leq \mu_J^{-s} \|u\|_{L_2} \|v\|_{\bar{\mathcal{H}}^s},$$

*and the induced norm $\| \cdot \|_\sharp^2 := \langle \cdot, \cdot \rangle_\sharp$ is equivalent to the $L_2$ norm.*

The weights $\mu_j$ and $\bar{\mathcal{H}}^s$ define a smoothness space, which is left generic for now but will be replaced by Sobolev spaces below.

*Proof.* We have

$$\left| \sum_{j=1}^{\infty} \left[ \lambda_j^2 - \bar{\lambda}_j^2 \right] \langle u, \psi_j \rangle \langle \psi_j, v \rangle \right| = \left| \sum_{j>J} \left[ \lambda_j^2 - \bar{\lambda}_j^2 \right] \langle u, \psi_j \rangle \langle \psi_j, v \rangle \right|$$

$$\leq \left[ \sum_{j>J} \left| \lambda_j^2 - \bar{\lambda}_j^2 \right| \langle u, \psi_j \rangle^2 \right]^{1/2} \left[ \sum_{j>J} \left| \lambda_j^2 - \bar{\lambda}_j^2 \right| \langle v, \psi_j \rangle^2 \right]^{1/2}$$

$$\leq \left[ \sum_{j>J} \langle u, \psi_j \rangle^2 \right]^{1/2} \left[ \sum_{j>J} \langle v, \psi_j \rangle^2 \right]^{1/2}$$

$$\leq \mu_J^{-s} \|u\|_{L_2} \|v\|_{\bar{\mathcal{H}}^s},$$

where the first equality follows from $\lambda_j = \bar{\lambda}_j$ for $j \leq J$, the second from Cauchy-Schwarz and the third from $0 \leq \bar{\lambda}_j^2 - \lambda_j^2 \leq 1$ by definition of $\bar{\lambda}_j$ for $j > J$. The last inequality follows from

$$\sum_{j>J} \langle u, \psi_j \rangle^2 \leq \|u\|_{L_2}^2$$

and

$$\sum_{j>J} \langle u, \psi_j \rangle^2 = \mu_J^{-2s} \sum_{j>J} \mu_J^{2s} \langle u, \psi_j \rangle^2 \leq \mu_J^{-2s} \sum_{j>J} \mu_j^{2s} \langle u, \psi_j \rangle^2 = \mu_J^{-2s} \|u\|_{\bar{\mathcal{H}}^s}^2 .$$

Finally, by construction, the eigenvalues $\bar{\lambda}_j \sim 1$ are upper and lower bounded by one and therefore the norm $\| \cdot \|_\sharp$ is equivalent to the $L_2$ norm. $\square$

### C.2 Concentration

We show concentration for

$$\sum_{r=1}^{R} \langle \kappa, \partial_r f_{\bar{\theta}} \rangle_{H^s} \left[ \partial_r \ell^k(\theta) - \partial_r \bar{\ell}^k(\theta) \right]$$

which matches Assumption (33), except for the wrong expected loss $\bar{\ell}^k(\theta)$ instead of $\ell_0(\theta)$, which will be considered in the next section. Throughout this section, we denote the adjoint of $u$, by $u^*$, so that $\langle f, u \rangle = f^* v$. For weights $\theta, \bar{\theta} \in \Theta$, we use the abbreviation

$$F_{\theta,\bar{\theta}}^s \kappa := \sum_{r=1}^{R} \partial_r f_\theta \langle \partial_r f_{\bar{\theta}}, \kappa \rangle_{H^s} \tag{43}$$

which shows up repeatedly in the following proofs. We first bound its norm.

**Lemma C.3.** *Assume that $\sigma$ satisfies the growth and Lipschitz conditions* (35), (36) *and may be different in each layer. Assume all weights in the networks, $\theta$, $\bar{\theta}$ and the domain are bounded* (38). *Then for* $0 \le S \le s < 1$

$$\|F_{\theta,\bar{\theta}}^S \kappa\|_{H^s} \le \|\kappa\|_{H^s}.$$

*Proof.* Let $\langle \cdot, \cdot \rangle$ be the $H^{-S} \times H^S$ dual pairing and $R : H^S \to H^{-S}$ the corresponding Riesz map. Since the $H^s$-norm is the dual of $H^{-s}$, we have

$$\|F_{\theta,\bar{\theta}}^S \kappa\|_{H^s} = \left\| \sum_{r=1}^{R} \partial_r f_\theta \langle \partial_r f_{\bar{\theta}}, \kappa \rangle_{H^S} \right\|_{H^s}$$

$$= \sup_{\|v\|_{H^{-s}} \le 1} \left\langle v, \sum_{r=1}^{R} \partial_r f_\theta \langle \partial_r f_{\bar{\theta}}, \kappa \rangle_{H^S} \right\rangle = \sup_{\|v\|_{H^{-s}} \le 1} \left\langle v, \left( \sum_{r=1}^{R} \partial_r f_\theta \partial_r f_{\bar{\theta}}^* \right) R\kappa \right\rangle$$

where in the last step we have used the definition of the adjoint functional $f_{\bar{\theta}}^* \in H^{-S}$ so that $\langle f_{\bar{\theta}}, \kappa \rangle_{H^S} = \langle f_{\bar{\theta}}, R\kappa \rangle = f_{\bar{\theta}}^*(R\kappa)$. Formally, in this argument $\langle \cdot, \cdot \rangle$ is the $H^S \times H^{-S}$ dual pairing, which reduces to the $L_2$ inner product in case $R\kappa$ is in $L_2$, see the proof of Lemma B.1 for more details. By Lemma D.2 cited from Welper (2024b), with $\epsilon$ sufficiently small so that $s + \epsilon < 1$, we obtain

$$\|F_{\theta,\bar{\theta}}^s \kappa\|_{H^s} \lesssim \sup_{\|v\|_{H^{-s}} \le 1} \|v\|_{H^{-s}} \|R\kappa\|_{H^{-s}} \le \|R\kappa\|_{H^{-s}} \le \|R\kappa\|_{H^{-S}} = \|\kappa\|_{H^S} \le \|\kappa\|_{H^s},$$

where we have used the embeddings $\| \cdot \|_{H^{-s}} \le \| \cdot \|_{H^{-S}}$ and $\| \cdot \|_{H^S} \le \| \cdot \|_{H^s}$ because $S \le s$.

$\qquad \square$

The next lemma shows the main concentration result.

**Lemma C.4.** *Assume that $\sigma$ satisfies the growth and Lipschitz conditions* (35), (36) *and may be different in each layer. Assume all weights in the networks, $\theta$, $\bar{\theta}$ and the domain are bounded* (38). *Assume the kernel is bounded by* $\sup_{x \in D} \|k(x, \cdot)\|_{L_2} \le C_k$. *Then for* $0 \le s < 1$ *with probability at least* $1 - 2t \left( e^t - t - 1 \right)^{-1}$ *we have*

$$\sum_{r=1}^{R} \langle \kappa, \partial_r f_{\bar{\theta}} \rangle_{H^s} \left[ \partial_r \ell^k(\theta) - \partial_r \bar{\ell}^k(\theta) \right] \lesssim C_k^2 \left( \sqrt{\frac{t}{N}} + \frac{t}{N} \right) \|\kappa\|_{L_2} \|\kappa\|_{H^s}.$$

*for all $\kappa \in H^s$.*

*Proof.* The lemma states that the gradient of the sample loss is close to its average, with high probability, which we show by the matrix Bernstein inequality. To this end, we first unravel the definition of the loss to

define appropriate random matrices:

$$\sum_{r=1}^{R} \langle \kappa, \partial_r f_{\bar{\theta}} \rangle_{H^s} \partial_r \ell^k(\theta) = \sum_{r=1}^{R} \langle \kappa, \partial_r f_{\theta} \rangle_{H^s} \left( \frac{1}{N} \sum_{i=1}^{N} \langle k(x_i, \cdot), \kappa \rangle \langle k(x_i, \cdot), \partial_r f_{\theta} \rangle \right)$$

$$= \frac{1}{N} \sum_{i=1}^{N} \sum_{r=1}^{R} \langle \kappa, k(x_i, \cdot) \rangle \langle k(x_i, \cdot), \partial_r f_{\theta} \rangle \langle \kappa, \partial_r f_{\bar{\theta}} \rangle_{H^s}$$

$$= \left\langle \kappa, \frac{1}{N} \sum_{i=1}^{N} k(x_i, \cdot) k(x_i, \cdot)^* \sum_{r=1}^{R} \partial_r f_{\theta} \langle \kappa, \partial_r f_{\bar{\theta}} \rangle_{H^s} \right\rangle$$

$$= \left\langle \kappa, M F_{\theta, \bar{\theta}}^s \kappa \right\rangle$$

with $F_{\theta, \bar{\theta}}^s \kappa$ defined in (43) and the integral kernel

$$M := \frac{1}{N} \sum_{i=1}^{N} k(x_i, \cdot) k(x_i, \cdot)^*.$$

With an analogous computation, we obtain the expectation with respect to the samples $x_i$:

$$\mathbb{E}\left[ \sum_{r=1}^{R} \langle \kappa, \partial_r f_{\bar{\theta}} \rangle_{H^s} \partial_r \ell^k(\theta) \right] = \left\langle \kappa, \bar{M} F_{\theta, \bar{\theta}}^s \kappa \right\rangle,$$

with

$$\bar{M} := \mathbb{E}\left[ \frac{1}{N} \sum_{i=1}^{N} k(x_i, \cdot) k(x_i, \cdot)^* \right] = \int_D k(x, \cdot) k(x_i, \cdot)^* \, dx.$$

It follows that

$$\sum_{r=1}^{R} \langle \kappa, \partial_r f_{\bar{\theta}} \rangle_{H^s} \left[ \partial_r \ell^k(\theta) - \partial_r \bar{\ell}^k(\theta) \right] = \left\langle \kappa, [M - \bar{M}] F_{\theta, \bar{\theta}}^s \kappa \right\rangle \leq \|\kappa\|_{L_2} \|M - \bar{M}\|_{H^s \to L_2} \|F_{\theta, \bar{\theta}}^s \kappa\|_{H^s} \quad (44)$$

The last term

$$\|F_{\theta, \bar{\theta}}^s \kappa\|_{H^s} \lesssim \|\kappa\|_{H^s} \quad (45)$$

is bounded by Lemma C.3 so that it remains to bound $\|M - \bar{M}\|_{H^s \to L_2}$. To this end note that $M$ is a sum of rank one operators, each depending on one independent sample $x_i$. Hence, concentration follows from dimension free matrix Bernstein inequalities (Hsu et al., 2012; Tropp, 2015; Minsker, 2017). We use a corollary shown in Gentile & Welper (2022) and stated in the supplementary material Lemma D.4 for which we only need to bound

$$\|k(x, \cdot)^*\|_{(H^s)^*} = \|k(x, \cdot)\|_{H^{-s}} \leq \|k(x, \cdot)\|_{L_2} \leq C_k$$

for all $x \in D$, provided by the assumptions of the lemma. With Lemma D.4 this provides

$$\Pr\left[ \|M - \bar{M}\|_{H^s \to L_2} \gtrsim C_k^2 \left( \sqrt{\frac{t}{N}} + \frac{t}{N} \right) \right] \leq 2t \left( e^t - t - 1 \right)^{-1}.$$

Together with (44) and (45) this proves the lemma.

$\square$

## C.3   Convergence: Proof of Theorem 2.4

We first bound $\Delta_{sample}(m, N)$ based on the results in the last two sections.

**Lemma C.5.** *Assume that $\sigma$ satisfies the growth and Lipschitz conditions* (35), (36) *and may be different in each layer. Assume all weights in the networks, $\theta$, $\bar\theta$ and the domain are bounded* (38). *Assume the kernel $k\colon D \times D \to \mathbb{R}$ is zonal, i.e. $k(x,y) = k(x^T y)$, and has eigenvalues $\lambda_{lj}$ with*

$$1 \lesssim \lambda_{lj} \lesssim 1, \quad l \leq L, \quad 1 \leq j \leq \nu(l),$$
$$\lambda_{lj} \lesssim 1, \quad l > L, \quad 1 \leq j \leq \nu(l)$$

*and index structure and $\nu(l)$ matching spherical harmonics* (6) *and*

$$\sup_{x \in D} \|k(x, \cdot)\|_{L_2} \leq C_k.$$

*Then for $0 \leq S \leq s < 1$ with probability at least $1 - 2t\left[e^t - t - 1\right]^{-1}$ for all $\kappa \in H^s$ we have*

$$\sum_{r=1}^{R} \langle \kappa, \partial_r f_{\bar\theta} \rangle_{H^S} \left[\partial_r \ell^k(\theta) - \partial_r \bar\ell^\sharp(\theta)\right] \lesssim C_k^2 \left[\sqrt{\frac{t}{N}} + \frac{t}{N}\right] \|\kappa\|_{L_2} \|\kappa\|_{H^S} + L^{-s} \|\kappa\|_{L_2} \|\kappa\|_{H^s},$$

*with loss $\bar\ell^\sharp(\theta)$ defined in* (42).

Note that the conclusion of the lemma contains both $S$ and $s$. This allows $S = s$, but also the choice $S = 0$ for which the last summand in the right hand side does not provide a meaningful bound if we insist that $S = s = 0$.

*Proof.* We first split the difference into expectation and concentration components:

$$\sum_{r=1}^{R} \langle \kappa, \partial_r f_{\bar\theta} \rangle_{H^S} \left[\partial_r \ell^k(\theta) - \partial_r \bar\ell^\sharp(\theta)\right]$$
$$= \sum_{r=1}^{R} \langle \kappa, \partial_r f_{\bar\theta} \rangle_{H^S} \left[\partial_r \ell^k(\theta) - \partial_r \bar\ell^k(\theta)\right] + \sum_{r=1}^{R} \langle \kappa, \partial_r f_{\bar\theta} \rangle_{H^S} \left[\partial_r \bar\ell^k(\theta) - \partial_r \bar\ell^\sharp(\theta)\right]$$
$$:= (I) + (II).$$

The first part $(I)$ is bounded by Lemma C.4:

$$(I) \lesssim C_k^2 \left(\sqrt{\frac{t}{N}} + \frac{t}{N}\right) \|\kappa\|_{L_2} \|\kappa\|_{H^S},$$

with probability at least $1 - 2t\left(e^t - t - 1\right)^{-1}$.

To estimate the expectation part $(II)$, first note that the Funk-Hecke formula (Atkinson & Han, 2012) implies that the eigenfunctions of the zonal kernel $k(\cdot, \cdot)$ are spherical harmonics $Y_l^j$ and thus $L_2$ orthogonal. Hence, we obtain the loss

$$\bar\ell^\sharp(\theta) := \frac{1}{2}\|f_\theta - f\|_\sharp^2 := \frac{1}{2} \sum_{l,j} \bar\lambda_{lj}^2 \left\langle Y_l^j, f_\theta - f \right\rangle^2, \qquad \bar\lambda_j := \begin{cases} \lambda_j & j \leq J, \\ \max\{\lambda_j, 1\} & j \geq J, \end{cases}$$

by Lemma C.2. The partial derivatives of the modified and expected kernel loss are given by

$$\partial_r \bar\ell^\sharp(\theta) = \sum_{l,j} \bar\lambda_{lj}^2 \left\langle \kappa, Y_l^j \right\rangle \left\langle Y_l^j, \partial_r f_\theta \right\rangle,$$

and

$$\partial_r \bar\ell^k(\theta) = \mathbb{E}\left[\frac{1}{N} \sum_{i=1}^{N} \langle \kappa, k(x_i, \cdot) \rangle \langle k(x_i, \cdot), \partial_r f_\theta \rangle\right] = \sum_{l,j} \lambda_{lj}^2 \left\langle \kappa, Y_l^j \right\rangle \left\langle Y_l^j, \partial_r f_\theta \right\rangle,$$

by Lemma C.1, respectively. It follows that

$$
\begin{aligned}
(II) &= \sum_{r=1}^{R} \langle \kappa, \partial_r f_{\bar{\theta}} \rangle_{H^s} \left[ \sum_{l,j} \left[ \lambda_{lj}^2 - \bar{\lambda}_{lj}^2 \right] \langle \kappa, Y_l^j \rangle \langle Y_l^j, \partial_r f_\theta \rangle \right] \\
&= \sum_{l,j} \left[ \lambda_{lj}^2 - \bar{\lambda}_{lj}^2 \right] \langle \kappa, Y_l^j \rangle \left\langle Y_l^j, \partial_r f_\theta \sum_{r=1}^{R} \langle \kappa, \partial_r f_{\bar{\theta}} \rangle_{H^s} \right\rangle \\
&= \sum_{l,j} \left[ \lambda_{lj}^2 - \bar{\lambda}_{lj}^2 \right] \langle \kappa, Y_l^j \rangle \left\langle Y_l^j, F_{\theta,\bar{\theta}}^S \kappa \right\rangle,
\end{aligned}
$$

with $F_{\theta,\bar{\theta}}^S \kappa$ defined in (43). Bounding the latter with Lemma C.3 and using Lemma C.2, we conclude that

$$
(II) \lesssim \mu_L^{-s} \|\kappa\|_{L_2} \|F_{\theta,\bar{\theta}}^S \kappa\|_{H^s} \lesssim \mu_L^{-s} \|\kappa\|_{L_2} \|\kappa\|_{H^s},
$$

where $\mu_l$ are the weights in the definition of the Sobolev space $H^s$ via spherical harmonics (5) and therefore $\mu_l \sim l$. Together with the bounds for $(I)$ this completes the proof.

$\square$

*Proof of Theorem 2.4.* The theorem follows from Theorem A.1. It shows gradient descent convergence in arbitrary scales of Hilbert spaces, for which the natural choice is $\mathcal{H}^s = H^s$. However, we replace the $L_2 = H^0$ norm with the equivalent $\| \cdot \|_\sharp$ norm to utilize our concentration result in Lemma C.5. This choice does not alter any assumptions or conclusions, except coercivity which uses the $\mathcal{H}^0$ inner product instead of the corresponding norm. We show that coercivity of the $L_2$ inner product implies coercivity of the $\| \cdot \|_\sharp$ inner product as well as the sample assumption (33). All other assumptions are proven in Welper (2024b, Theorem 2.2) for an analogous result without sample error.

1. *Coercivity* (28)*:* Since the neural tangent kernel is zonal, by the Funk-Hecke formula (Atkinson & Han, 2012), it has spherical harmonics as eigenfunctions with some eigenvalues $\mu_{lj}$, so that we have

$$
\langle v, Hv \rangle_\sharp = \sum_{l,j} \bar{\lambda}_l^2 \mu_{lj} v_{lj}^2 \sim \sum_{l,j} \mu_{lj} v_{lj}^2 = \langle v, Hv \rangle,
$$

with $v_{lj} = \langle v, Y_l^j \rangle$ and the $\| \cdot \|_\sharp$ norm from (42). Hence, $\| \cdot \|_\sharp$-coercivity is equivalent to the regular $L_2$-coercivity.

2. *Assumption* (33)*:* We show that with high probability for all $\theta, \bar{\theta}$ with $\|\cdot\|_*$ distance to the initial $\theta^0$ smaller than $\lesssim h$ and $S \in \{0, s\}$ we have the bound

$$
\sum_{r=1}^{R} \langle \kappa, \partial_r f_{\bar{\theta}} \rangle_{H^S} \left[ \partial_r \ell^k(\theta) - \ell_\sharp(\theta) \right] \le \Delta_{sample}(m, N) \|\kappa\|_{L_2} \|\kappa\|_{H^S} \tag{46}
$$

with

$$
\Delta_{sample}(m, N) \lesssim C_k^2 \left( \frac{\tau_N}{N} \right)^{1/2} + C_k^{-2} \left( \frac{N}{\tau_N} \right)^{1/2} L^{-s} + L^{-s}, \tag{47}
$$

which provides Assumption (33). To this end, by Lemma C.5 with $t = \tau_N$ and probability at least $1 - 2\tau_N \left[ e^{\tau_N} - \tau_N - 1 \right]^{-1}$ for all $\kappa \in \mathcal{H}^s$ we have

$$
\sum_{r=1}^{R} \langle \kappa, \partial_r f_{\bar{\theta}} \rangle_{H^S} \left[ \partial_r \ell^k(\theta) - \bar{\ell}^\sharp(\theta) \right] \lesssim C_k^2 \left( \frac{\tau_N}{N} \right)^{1/2} \|\kappa\|_{L_2} \|\kappa\|_{H^S} + L^{-s} \|\kappa\|_{L_2} \|\kappa\|_{H^s}. \tag{48}
$$

This yields the claimed bounds (46), (47) for $S = s$. For $S = 0$ the last summand has the wrong norm: $L^{-s} \|\kappa\|_{L_2} \|\kappa\|_{H^s}$ instead of $L^{-s} \|\kappa\|_{L_2} \|\kappa\|_{H^S} = L^{-s} \|\kappa\|_{L_2}^2$. To replace the $H^s$ norm with an $L_2$ norm note that assumption (34) yields

$$\|\kappa^k\|_{L_2}^2 \geq c_a \left( m^{-\frac{1}{2}\frac{\alpha}{1+\alpha}} + \Delta_{sample}(m,N) \right)^{\frac{s}{\beta}} \|\kappa^0\|_{H^s}^2 \gtrsim C_k^2 \left( \frac{\tau_N}{N} \right)^{1/2} \|\kappa^0\|_{H^s}^2$$

for $k \leq n$. Moreover, by induction, from Theorem A.1 we have $\|\kappa^k\|_{H^s} \lesssim \|\kappa^0\|_{H^s}$ and therefore arrive at

$$\|\kappa^k\|_{H^s}^2 \lesssim C_k^{-2} \left( \frac{N}{\tau_N} \right)^{1/2} \|\kappa^k\|_{L_2}.$$

Together with (48) this yields the claimed bounds (46) and (47) for the case $S = 0$. Together with the case $S = s$ above, this establishes assumption (33).

This completes the proof, together with an index shift $n + 1 \to n$ between the statements of Theorems A.1 and 2.2.

$\square$

# D   Supplementary Material

## D.1   Technical Lemmas

**Lemma D.1.** *Let $k_t(x, y)$ be the heat kernel defined in* (21). *Then for all $y \in \mathbb{S}^{d-1}$*

$$\|k_t(\cdot, y)\|_{H^s(\mathbb{S}^{d-1})}^2 \lesssim t^{-s-d+3/2}.$$

*Proof.* Plugging the definition of the heat kernel (21) into the definition of Sobolev norms (5), we obtain

$$\|k_t(\cdot, y)\|_{H^s(\mathbb{S}^{d-1})}^2 = \sum_{l=0}^{\infty} \sum_{j=1}^{\nu(l)} \left( 1 + l^{1/2}(l+d-2)^{1/2} \right)^{2s} \left| e^{-l(l+d-2)t} Y_l^j(y) \right|^2$$

$$\lesssim 1 + \sum_{l=0}^{\infty} l^{2s} e^{-2l^2 t} \sum_{j=1}^{\nu(l)} \left| Y_l^j(y) \right|^2.$$

Since $|Y_l^j(y)|^2 \lesssim \nu(l)$ and $\nu(l) \lesssim l^{d-2}$, see Stein & Weiss (1972, Chapter 4.2, Corollary 2.9), we obtain

$$\|k_t(\cdot, y)\|_{H^s(\mathbb{S}^{d-1})}^2 \lesssim 1 + \sum_{l=0}^{\infty} l^{2s} l^{2d-4} e^{-2l^2 t} \lesssim 1 + \int_0^{\infty} l^{2s+2d-4} e^{-2l^2 t}\, dl$$

$$= 1 + t^{-s-d+3/2} \int_0^{\infty} x^{2s+2d-4} e^{-2x^2}\, dx \lesssim t^{-s-d+3/2},$$

were we have substituted $x = l\sqrt{t}$ and used that the latter integral is bounded.

$\square$

## D.2   Results from Gentile & Welper (2022); Welper (2024b;a)

To keep the paper self contained, this section contains several results from Gentile & Welper (2022); Welper (2024b;a).

**Lemma D.2.** *Assume that $\sigma$ and $\dot{\sigma}$ satisfy the growth and Lipschitz conditions* (35), (36) *and may be different in each layer. Assume the weights, perturbed weights and domain are bounded* (38) *and $m_L \sim m_{L-1} \sim \cdots \sim m_1$. Then for $0 < s < 1$ and $m_0 := m_1$*

$$\iint_{D \times D} f(x) \left( \sum_{r=1}^{R} \partial_r f_\theta(x) \partial_r f_\theta(y) \right) g(y)\, dx\, dy \lesssim \|f\|_{H^{-s}(\mathbb{S}^{d-1})} \|g\|_{H^{-s}(\mathbb{S}^{d-1})}.$$

*Proof.* This is a direct consequence of Welper (2024b, Lemma 7.16) and Welper (2024b, Lemma 6.7). For $\epsilon$ sufficiently small so that $s + \epsilon < 1$, the former shows that

$$\iint_{D \times D} f(x) k(x,y) g(y) \, dx \, dy \leq \|f\|_{H^{-s}(\mathbb{S}^{d-1})} \|g\|_{H^{-s}(\mathbb{S}^{d-1})} \|k\|_{C^{0;s+\epsilon,s+\epsilon}(\mathbb{S}^{d-1})},$$

for $k(x,y) = \sum_{r=1}^{R} \partial_r f_\theta(x) \partial_r f_\theta(y)$ and where $\|\cdot\|_{C^{0;s+\epsilon,s+\epsilon}(\mathbb{S}^{d-1})}$ is a Hölder norm of order $s + \epsilon$ in the two variables $x$ and $y$. Technically, the reference does not include the case $s = 0$, which follows directly from a sup-norm bound and the fact that the domain is bounded. The second reference Welper (2024b, Lemma 6.7) shows that

$$\|k\|_{C^{0;s+\epsilon,s+\epsilon}} \lesssim 1.$$

where $k(x,y) = \sum_{r=1}^{R} \partial_r f_\theta(x) \partial_r f_\theta(y)$ is denoted by $\bar{\bar{\Gamma}}$. Combining the two inequalities yields the result.

$\square$

**Lemma D.3** (Welper (2024b, Lemma 6.2))**.** *Assume that* $\|x\| \lesssim 1$.

1. *Assume that* $\sigma$ *satisfies the growth condition* (35) *and may be different in each layer. Assume the weights are bounded* (38)*. Then*

$$\left\| f^\ell(x) \right\| \lesssim m_0^{1/2} \prod_{k=0}^{\ell-1} \left\| W^k \right\| m_k^{-1/2}.$$

2. *Assume that* $\sigma$ *satisfies the growth and Lipschitz conditions* (35) *and* (36) *and may be different in each layer. Assume the weights and perturbed weights are bounded* (38)*. Then*

$$\left\| f^\ell(x) - \bar{f}^\ell(x) \right\| \lesssim m_0^{1/2} \sum_{k=0}^{\ell-1} \left\| W^k - \bar{W}^k \right\| m_k^{-1/2} \prod_{\substack{j=0 \\ j \neq k}}^{\ell-1} \max \left\{ \left\| W^j \right\|, \left\| \bar{W}^j \right\| \right\} m_j^{-1/2}.$$

**Lemma D.4** (Gentile & Welper (2022, Corollary 6.4))**.** *Let* $\xi_i$, $i = 1, \dots, n$ *be independent random variables,* $U$, $V$ *Hilbert spaces and* $X_i = X_i(\xi_i) = v_i(\xi_i) u_i(\xi_i)^* = v_i u_i^*$ *be Bochner integrable rank one operators with* $v_i \in V$ *and* $u_i^* \in U^*$*. Assume there are* $\mu > 0$ *and* $\nu > 0$ *such that for all* $i = 1, \dots, n$

$$\|u^*\|_{U^*} \leq \mu, \qquad\qquad \|v\|_V \leq \nu,$$

*almost surely. Then, for any* $t > 0$,

$$\Pr \left[ \left\| \frac{1}{n} \sum_{i=1}^{n} X_i - \mathbb{E}[X_i] \right\| > \sqrt{\frac{8\mu^2 \nu^2 t}{n}} + \frac{2\mu\nu t}{3n} \right] \leq 2t \left( e^t - t - 1 \right)^{-1}.$$

*Proof.* The only difference to the reference is that we assume $\|u^*\|_{U^*} \leq \mu$ instead of $\|u\|_U \leq \mu$, which is equivalent due to the Riesz representation theorem. $\square$

**Lemma D.5** (Welper (2024a, Lemma 3.3))**.** *Let* $a, b, c, d > 0$ *and* $\rho > 1/2$*. Let* $u_n$ *and* $v_n$ *be two sequences that satisfy*

$$\begin{aligned} u_{n+1} - u_n &\leq -\gamma a u_n^{1+\rho} v_n^{-\rho} + \gamma b u_n, \\ v_{n+1} - v_n &\leq -\gamma c u_n^\rho v_n^{1-\rho} + \gamma d \sqrt{u_n v_n}. \end{aligned} \tag{49}$$

*Furthermore, assume that*

$$u_k \geq \left( \frac{d}{c} \right)^{\frac{2}{2\rho-1}} v_0, \qquad u_k \geq \left( 2\frac{b}{a} \right)^{\frac{1}{\rho}} v_0, \qquad \text{for all } k = 0, \dots, n-1. \tag{50}$$

*Then*

$$u_n \leq e^{-\gamma bn} u_0, \qquad\qquad v_n \leq v_0.$$

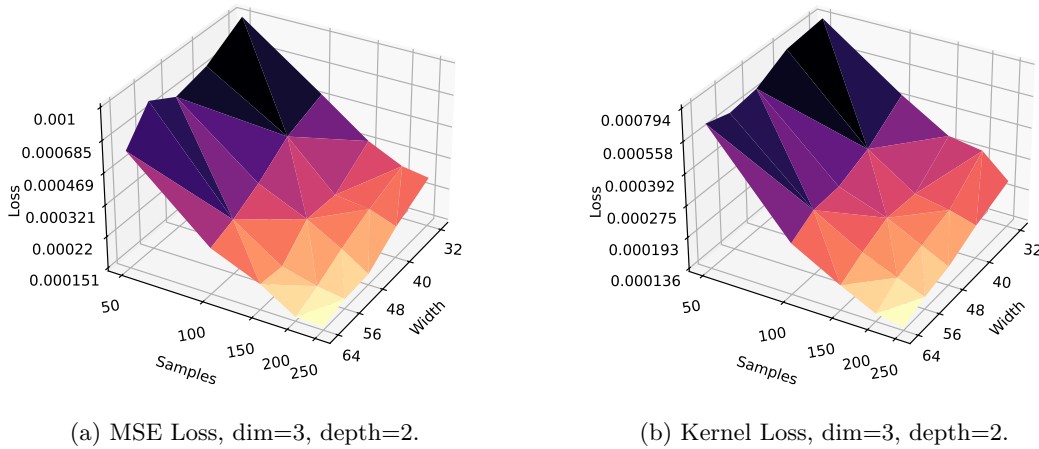

(a) MSE Loss, dim=3, depth=2.    (b) Kernel Loss, dim=3, depth=2.

Figure 2: Test loss for training with mean squared loss (13) (left) and (17) (right). All axes are log-scaled so that the slope corresponds to convergence rates.

## E   Extra Numerical Experiments

This appendix contains some extra numerical results for the setup in Section 3.

- Section 3 does not report any losses. They are contained in the extended Table 2.

- This section also contains experiments for shallow networks in three dimensions, shown in Figure 2 and Table 2.

- Figure 3 contains results for the shallow network with MSE loss in 7 dimensions and with larger range for samples and width.

The observations from Section 3 remain unchanged. The deep networks performs slightly better than the shallow ones, but have significantly more weights in total.

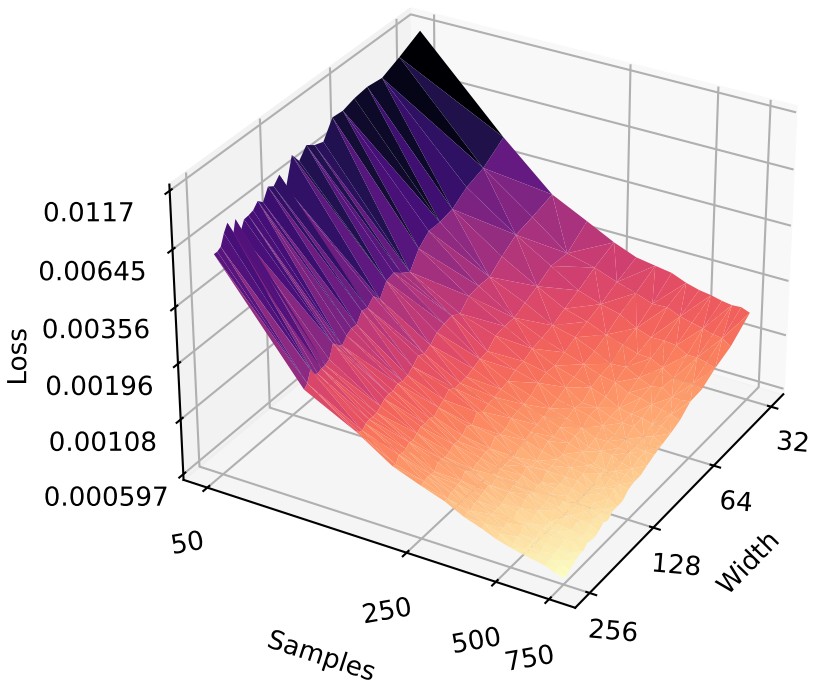

Figure 3: Test loss for training with mean squared loss (13) for dimension 7 and depth 2. All axes are log-scaled so that the slope corresponds to convergence rates.

### Dimension 3 and Depth 2 – Trained with MSE Loss

| m/N | Test Loss | | | | dof rate | | | | N rate | | | |
|---|---|---|---|---|---|---|---|---|---|---|---|---|
| | 100 | 150 | 200 | 250 | 100 | 150 | 200 | 250 | 100 | 150 | 200 | 250 |
| 40 | 0.00043 | 0.00025 | 0.000238 | 0.000212 | 0.719 | 1.3 | 0.659 | 0.955 | 0.832 | 1.34 | 0.169 | 0.524 |
| 48 | 0.00033 | 0.000256 | 0.000203 | 0.000165 | 1.45 | -0.119 | 0.883 | 1.36 | 1.08 | 0.631 | 0.805 | 0.917 |
| 56 | 0.000279 | 0.00019 | 0.000171 | 0.000163 | 1.1 | 1.92 | 1.1 | 0.0938 | 1.6 | 0.943 | 0.367 | 0.222 |
| 64 | 0.000252 | 0.000197 | 0.000153 | 0.000151 | 0.768 | -0.273 | 0.859 | 0.58 | 1.22 | 0.601 | 0.893 | 0.0546 |

### Dimension 3 and Depth 5 – Trained with MSE Loss

| m/N | Test Loss | | | | dof rate | | | | N rate | | | |
|---|---|---|---|---|---|---|---|---|---|---|---|---|
| | 100 | 150 | 200 | 250 | 100 | 150 | 200 | 250 | 100 | 150 | 200 | 250 |
| 40 | 0.000268 | 0.000155 | 0.000125 | 0.000102 | 0.331 | 1.33 | 1.33 | 1.19 | 1.9 | 1.36 | 0.743 | 0.896 |
| 48 | 0.00023 | 0.000146 | 0.000111 | 8.44e-05 | 0.833 | 0.328 | 0.628 | 1.06 | 1.86 | 1.13 | 0.933 | 1.25 |
| 56 | 0.000194 | 0.000126 | 9.19e-05 | 7.89e-05 | 1.13 | 0.969 | 1.25 | 0.434 | 2.2 | 1.07 | 1.08 | 0.684 |
| 64 | 0.000222 | 9.55e-05 | 8.08e-05 | 6.49e-05 | -1.03 | 2.05 | 0.967 | 1.47 | 1.56 | 2.08 | 0.582 | 0.985 |

### Dimension 3 and Depth 2 – Trained with Kernel Loss

| m/N | Test Loss | | | | dof rate | | | | N rate | | | |
|---|---|---|---|---|---|---|---|---|---|---|---|---|
| | 100 | 150 | 200 | 250 | 100 | 150 | 200 | 250 | 100 | 150 | 200 | 250 |
| 40 | 0.000317 | 0.000223 | 0.000208 | 0.00019 | 1.29 | 1.43 | 1.47 | 0.677 | 1.26 | 0.871 | 0.246 | 0.388 |
| 48 | 0.000275 | 0.000227 | 0.000187 | 0.000162 | 0.784 | -0.11 | 0.563 | 0.898 | 1.2 | 0.469 | 0.672 | 0.662 |
| 56 | 0.000271 | 0.000199 | 0.000169 | 0.00015 | 0.103 | 0.858 | 0.658 | 0.474 | 1.16 | 0.756 | 0.565 | 0.536 |
| 64 | 0.000233 | 0.000173 | 0.000143 | 0.000136 | 1.12 | 1.05 | 1.26 | 0.756 | 1.49 | 0.732 | 0.664 | 0.232 |

### Dimension 3 and Depth 5 – Trained with Kernel Loss

| m/N | Test Loss | | | | dof rate | | | | N rate | | | |
|---|---|---|---|---|---|---|---|---|---|---|---|---|
| | 100 | 150 | 200 | 250 | 100 | 150 | 200 | 250 | 100 | 150 | 200 | 250 |
| 40 | 0.00019 | 0.00013 | 0.000103 | 9.29e-05 | 1.08 | 0.676 | 0.748 | 1.07 | 1.65 | 0.939 | 0.797 | 0.482 |
| 48 | 0.000183 | 0.000118 | 9.56e-05 | 7.67e-05 | 0.229 | 0.517 | 0.433 | 1.05 | 1.23 | 1.07 | 0.744 | 0.987 |
| 56 | 0.000141 | 9.01e-05 | 7.29e-05 | 6.47e-05 | 1.69 | 1.77 | 1.75 | 1.11 | 2.02 | 1.1 | 0.734 | 0.539 |
| 64 | 0.000151 | 9.27e-05 | 6.47e-05 | 5.52e-05 | -0.54 | -0.218 | 0.898 | 1.19 | 1.71 | 1.21 | 1.25 | 0.713 |

Table 2: Loss and estimated convergence rates between neighbouring losses for the given $m/N$. Left: Rate along the column, i.e. with respect to $m$. Right: Rate along rows, i.e. with respect to number of samples $N$. The first table is trained with mean squared loss (MSE) (13) and the second with kernel loss (17).

