# OpenReview forum: "Approximation, Estimation and Optimization Errors for a Deep Neural Network"
_TMLR — Accepted by TMLR_

### Review · Reviewer_gMn3 · 2025-03-12

**Summary Of Contributions:**

This paper addresses the challenge of analyzing all three main sources of error in neural network learning---namely approximation, estimation, and optimization errors. Typically, each error type is studied in isolation or with partial overlap. In contrast, this paper aims to provide a unified theoretical framework that bounds all three simultaneously under specific settings involving fully connected networks trained by gradient descent, examining both standard $L2$ loss and a kernel loss. In this setup, the training samples $x_i$ lie on the unit sphere, and $y_i = f(x_i)$ represents the function values that the network aims to interpolate. A key takeaway is that, under the neural tangent kernel (NTK) regime, while the network’s generalization error remains above a certain threshold, the gradient descent procedure reduces the error exponentially fast in each iteration.


I enjoyed reading the paper and I think the results presented in the paper deliver valuable insights. It would be very helpful if the authors could clarify the questions below.

**Audience:**

Yes

**Claims And Evidence:**

Yes

**Requested Changes:**

Requested Changes:

1. Equation (4) does not match Equation (16).
2. On page 5, there is a typo in the lines beginning with $ g_{\theta}(x)$; an extra “)” appears.
3. On page 5, what is the purpose of introducing $g(x)$ and $ g_{\theta}(x)$? It seems that setting $ g(x) \equiv 0 $ might be possible.
4. On page 6, under Equation (7), there are multiple typos in the lines beginning with $W^L$, $W^{L-1}$, and $ W^1 $. This makes it unclear which weight matrices are being trained and which are not.
5. On page 11, the word "permutation" should probably be replaced by "perturbation". The first equation containing the term PERMUTATION needs to be realigned.
6. On page 12, under "Remark on Generalization Errors and Optimization", the norms $||W^{L-1} - W_0^{L-1}||$ and $|| \theta-\theta_0 ||_*$ are introduced, but their definitions do not appear until page 28.
7. On page 26, the variables $x_{n+1}$ and $y_{n+1}$ are used as variables distinct from the training data $(x_i,y_i)$.

**Strengths And Weaknesses:**

Strengths:

1. This paper is well-written and easy to follow.
2. This paper provides a nice analysis of gradient descent in neural network training by systematically handling approximation, estimation, and optimization errors in one framework.
3. The analysis seems rigorous though I didn't check the proofs thoroughly.


Questions:

1. What is the benefit of assuming $x_i$ lies on the unit sphere? Is there a transformation allowing you to map a general $x_i \in \mathbb{R}^{d-1}$ to the unit sphere in $\mathbb{R}^d$?
For instance, using the transformation $  x \mapsto \frac{(x,1)}{\sqrt{||x||^2 + 1}},$ can the authors extend the current results to $x_i$ in $\mathbb{R}^{d-1}$?

2. If considering the case $y = f(x) + e$ instead of the exact interpolation problem, what is the main difficulty?

3. How does the hidden layer size $m$ affect Theorems 2.2 and 2.3, aside from influencing the upper bound on $||f||$ and the probability lower bound?

---

> ### Author Response · Authors · 2025-04-16
> **Reply**
>
> Thank you for the detailed review. All changes in the manuscript are in blue and comments to all items below.
>
> **Questions**
>
> 1. The networks do not have a bias and as a result, the input $x=0$ always yields output $f_\theta(x) = 0$, independent of the network weights. This degenerate case is avoided by restricting the input to the sphere. Introducing a bias would likely avoid this issue and allow more general domains, even without the transforms indicated by the reviewer. But then, we cannot use several cited results form the NTK literature (Appendix D), which also use a sphere.
>
> 1. The gradient descent convergence is shown for the expected loss. The sample or kernel losses introduce an extra error in the gradients, that are controlled during iteration. A noise term could be treated similarly, in which case it would show up in a bound like (32) and modify $\Delta_{sample}$ or induce a similar perturbation term.
>
> 1. All constants in the theorems are independent of the hidden layer size $m$. Simplifying and condensing them into $a, b, c \ge 0$, after gradient descent has finished training, one obtains error bounds of the form
>    $$
>      \|f_\theta - f\|_{L_2(\mathbb{S}^{d-1})}^2 \lesssim m^{-a} + m^{b} N^{-c},
>    $$
>    see also Section 1.2 and the new Remarks 2.3 and 2.5. The first summand converges to zero for $m \to \infty$. To control the second summand, we must have sufficiently many samples $N$. For the kernel loss, we have $b=0$ and obtain the worst case of approximation ($m$) and sample ($N$) error.
>
>    Some concrete numbers for the constants $a,b,c$ are given in Section 3 on numerical experiments.
>
> **Requested Changes**
>
> 1. Corrected equation (4).
> 1. Done.
> 1. The min should have been an argmax. Corrected.
> 1. Corrected.
> 1. Corrected. Replaced "permutation" throughout the text.
> 1. Added the definitions.
> 1. Changed the notation to $x_n \to u^n$ and $y_n \to v^n$.

---

### Review · Reviewer_2mbw · 2025-03-23

**Summary Of Contributions:**

This paper presents an NTK-based analysis of deep neural networks with only the second-to-last layer trained and proves that the test error could be bounded as $\lesssim m^{-a} + m^b N^{-c}$, where $m$ is the width and $N$ is the sample size. Two settings are considered: pointwise sampling and kernel sampling.

**Audience:**

Yes

**Broader Impact Concerns:**

/

**Claims And Evidence:**

Yes

**Requested Changes:**

I would like to see the authors' responses to my Weaknesses 1 - 3 raised above.

**Strengths And Weaknesses:**

Strengths:
1. The paper is well-written.
2. Settings and assumptions are stated clearly.
3. The new bound established in the paper requires less overparameterization, thus reducing the gap between theory and practice.

Weaknesses:
1. While the introduction states the bound as $\lesssim m^{-a} + m^b N^{-c}$, the main theorems 2.2 and 2.3 do not state the result in this way. I wonder if the authors could resolve this discrepancy.
2. In contrast to real-world neural network training, only the second-to-last layer is trained in the paper. I wonder if the main results still hold if more layers are trained.
3. The data is assumed to be sampled from the uniform distribution on a sphere, which is a very restrictive assumption. I wonder if the main results still hold if a more general class of distributions is considered.
4. While the bound may need less overparameterization than previous works, modern neural networks are certainly not trained in the NTK regime. A more substantial improvement over existing analyses would be analyzing the training regime beyond the NTK, even if this may not lead to a better test error bound / less overparameterization in the short term.

---

> ### Author Response · Authors · 2025-04-16
> **Reply**
>
> Thank you for the review. Changes in the manuscript are in blue and comments to each item in *Weaknesses* are below.
>
> 1. We've added Remarks 2.3 and 2.5, which show how the theorems yield the simplified statements in the introduction.
>
> 1. We conjecture that the main result still holds, but that it would require substantially more work: In the standard NTK argument, the convergence analysis of all layers can be reduced to the second but last layer by dropping positive (layer-wise) terms in the NTK. Since we also track the smoothness of gradient descent iterates, we presume that such a simplification works only approximately after careful concentration analysis of the NTK in smoothness norms. This is expected to be lengthy, non-trivial and not contained in the literature that we rely on (Appendix D).
>
> 1. For generalization from the sphere to more general domains, see the answer to Question 1 of Reviewer gMn3. Non-uniform distributions would likely require some changes. In particular, the Sobolev norms in the smoothness bounds would need to be adapted accordingly.
>
> 1. In our analysis the kernel regime does not originate from over-parametrization ($m \gg N$) but from limiting the training time in (14). We show that in this initial training phase, we already achieve favorable errors. If we keep training, the network may well leave the linearized regime and show nonlinear effects, which we agree would be very interesting to understand better.

---

### Review · Reviewer_naT1 · 2025-04-01

**Summary Of Contributions:**

This paper aims to provide a unified framework for quantifying approximation, estimation, and optimization errors when using gradient descent to minimize empirical loss. The authors establish theorems showing that the total error decreases exponentially until it reaches a level. The results apply to least squares loss in the under-parameterized regime, and to kernel-based losses in both under- and over-parameterized settings.

**Audience:**

Yes

**Broader Impact Concerns:**

N.A

**Claims And Evidence:**

Yes

**Requested Changes:**

1. In the equation above (2), authors use $\hat \theta$ on the left-hand side and $ \theta$ on the right-hand side.

2. Should "permutation" be "perturbation"?

**Strengths And Weaknesses:**

**Strengths:**

1. The paper offers a fresh perspective that differs from much of the existing literature.

2. It considers of both under- and over-parameterized regimes, without requiring different assumptions for analyzing approximation, estimation, and optimization errors.

**Weaknesses and Questions:**

1. While the theorem statements are clear, it remains unclear how the approximation, estimation, and optimization errors are individually characterized. For example, in Theorem 2.2, Equation (14) includes terms like $ m^{-1/2} $ and $ \Delta_{\text{sample}} $, which appear to represent approximation and estimation errors, respectively, since one diminish with $ m $ and the other diminishes with  $ m $  and $ N $. However, these do not seem to align directly with the definitions provided in the introduction. Does this mean the authors did not follow the error decomposition as defined earlier?

2. This leads to a broader concern: the remaining term in Equation (14) seems to be tied to the choice of algorithm. Specifically, the $ m^{-1/2} $ term arises due to the use of gradient descent, and $ \Delta_{\text{sample}} $ stems from perturbations analysis of gradient descent. However, the definition of approximation (and to some extent, estimation) error should be independent of the algorithm. Put differently, if we were to use an optimization method other than gradient descent, could we potentially obtain a better bound in Equation (14)?

---

> ### Author Response · Authors · 2025-04-16
> **Reply**
>
> Thank you for the questions and corrections. Changes in the manuscript are in blue and comments to each item below.
>
> **Weaknesses and Questions**
>
> 1. The error decomposition in the introduction considers all three error components separately, as usual in the literature. The main theorems do not because the proofs do not follow standard arguments. The $m^{1/2}$ and $\Delta_{sample}$ terms can be traced back to perturbations from approximation and sampling in an idealized gradient descent scheme.
>
>    Although not stated directly, the main theorems do imply individual approximation errors: The best approximation must be better than the one generated by gradient descent, for any number of samples $N$. In particular, for $N \to \infty$, the bound (14) reduces an upper bound for the approximation error of type (1).
>
> 1. For approximation, we can consider three different errors:
>
>    1. $\|f_\theta - f\|$, where $\theta$ is the global minimizer.
>    1. $\|f_\theta - f\|$, where $\theta$ is trained by a practical algorithm, e.g. gradient descent.
>    1. Provable upper bounds for the latter.
>
>    Note that any bound for the last item also bounds the first item and hence the approximation error as defined in the manuscript. If the algorithm does not find the global optima, the first and last error may well be different.
>
>    In this paper, we consider the last error, which does indeed depend on the algorithm. This is expected because the numerical experiments indicate that in the given setup gradient descent does not find the global optimum.
>
> **Requested Changes:**
>
> 1. Corrected.
> 1. Yes, corrected throughout the document.

---

### Comment · Reviewer_gMn3 · 2025-02-17

Summary Of Contributions:

This paper addresses the challenge of analyzing all three main sources of error in neural network learning---namely approximation, estimation, and optimization errors. Typically, each error type is studied in isolation or with partial overlap. In contrast, this paper aims to provide a unified theoretical framework that bounds all three simultaneously under specific settings involving fully connected networks trained by gradient descent, examining both standard $L2$ loss and a kernel loss. In this setup, the training samples $x_i$ lie on the unit sphere, and $y_i = f(x_i)$ represents the function values that the network aims to interpolate. A key takeaway is that, under the neural tangent kernel (NTK) regime, while the network’s generalization error remains above a certain threshold, the gradient descent procedure reduces the error exponentially fast in each iteration.

Strength:
1. This paper is well-written and easy to follow.
2. This paper provides a nice analysis of gradient descent in neural network training by systematically handling approximation, estimation, and optimization errors in one framework.
3. The analysis seems rigorous though I didn't check the proofs thoroughly.

Questions:

1. What is the benefit of assuming $x_i$ lies on the unit sphere? Is there a transformation allowing you to map a general $x_i \in \mathbb{R}^{d-1}$ to the unit sphere in $\mathbb{R}^d$?
For instance, using the transformation $  x \mapsto \frac{(x,1)}{\sqrt{||x||^2 + 1}},$ can the authors extend the current results to $x_i$ in $\mathbb{R}^{d-1}$?

2. If considering the case $y = f(x) + e$ instead of the exact interpolation problem, what is the main difficulty?

3. How does the hidden layer size $m$ affect Theorems 2.2 and 2.3, aside from influencing the upper bound on $||f||$ and the probability lower bound?

Requested Changes:

1. Equation (4) does not match Equation (16).
2. On page 5, there is a typo in the lines beginning with $ g_{\theta}(x)$; an extra “)” appears.
3. On page 5, what is the purpose of introducing $g(x)$ and $ g_{\theta}(x)$? It seems that setting $ g(x) \equiv 0 $ might be possible.
4. On page 6, under Equation (7), there are multiple typos in the lines beginning with $W^L$, $W^{L-1}$, and $ W^1 $. This makes it unclear which weight matrices are being trained and which are not.
5. On page 11, the word "permutation" should probably be replaced by "perturbation". The first equation containing the term PERMUTATION needs to be realigned.
6. On page 12, under "Remark on Generalization Errors and Optimization", the norms $||W^{L-1} - W_0^{L-1}||$ and $|| \theta-\theta_0 ||_*$ are introduced, but their definitions do not appear until page 28.
7. On page 26, the variables $x_{n+1}$ and $y_{n+1}$ are used as variables distinct from the training data $(x_i,y_i)$.


Summary Of The Review:

I enjoyed reading the paper and I think the results presented in the paper deliver valuable insights. It would be very helpful if the authors could clarify the questions above.

---

> ### Comment · Editors_In_Chief · 2025-03-12
>
> Reviewer, you posted this as a Comment -- can you please post this instead as a Review?
>
> Gautam

---

### Decision · Action_Editor_ADUK · 2025-06-22

**Recommendation:** Accept with minor revision

**Additional Comments:**

This paper has been substantially improved and the reviewers generally agree on acceptance. The key remaining issue is the strong assumptions in the theory: 1) only the second-to-last layer is trained and 2) the data is assumed to be sampled from the uniform distribution on a sphere. The authors discussed ways that (2) could be relaxed and that they hypothesize that (1) could be relaxed but it would much harder.

The decision is Accept with Minor Revisions, where I will check two minor revisions.
1. It is ok that these stronger assumptions are used, but this needs to be made more transparent early-on. The abstract makes a stronger claim about the scope of what is achieved. It should be added to the abstract and intro, highlighting that you achieve this novel result, but currently still under somewhat restrictive assumptions.
2. The authors should add their two comments on (1) and (2), that they gave to reviewers, briefly in a discussion or conclusion.

**Audience:**

Yes

**Audience Explanation:**

Understanding convergence properties of deep NNs is widely of interest. This paper has some restrictive assumptions, potentially narrowing the interest, but the novelty of the outcome is sufficient to interest some in the TMLR community and good provide ideas for expanding this analysis.

**Claims And Evidence:**

Yes

**Claims Explanation:**

The reviewers agreed that the paper's theoretical arguments are correct and now clearly laid out.